# Replication stress induces mitotic death through parallel pathways regulated by WAPL and telomere deprotection

V. Pragathi Masamsetti [1], Ronnie Ren Jie Low [1,2,3], Ka Sin Mak[1], Aisling O'Connor[1], Chris D. Riffkin[2], Noa Lamm[1], Laure Crabbe[4,5], Jan Karlseder[6], David C.S. Huang[2,3], Makoto T. Hayashi[7] & Anthony J. Cesare [1]

Mitotic catastrophe is a broad descriptor encompassing unclear mechanisms of cell death. Here we investigate replication stress-driven mitotic catastrophe in human cells and identify that replication stress principally induces mitotic death signalled through two independent pathways. In p53-compromised cells we find that lethal replication stress confers WAPL-dependent centromere cohesion defects that maintain spindle assembly checkpoint-dependent mitotic arrest in the same cell cycle. Mitotic arrest then drives cohesion fatigue and triggers mitotic death through a primary pathway of BAX/BAK-dependent apoptosis. Simultaneously, a secondary mitotic death pathway is engaged through non-canonical telomere deprotection, regulated by TRF2, Aurora B and ATM. Additionally, we find that suppressing mitotic death in replication stressed cells results in distinct cellular outcomes depending upon how cell death is averted. These data demonstrate how replication stress-induced mitotic catastrophe signals cell death with implications for cancer treatment and cancer genome evolution.

--------------------------------------------------------------------------------

[1] Children's Medical Research Institute, University of Sydney, Westmead, NSW 2145, Australia. [2] The Walter and Eliza Hall Institute of Medical Research, Parkville, VIC 3052, Australia. [3] Department of Medical Biology, The University of Melbourne, Parkville, VIC 3052, Australia. [4] Institute for Integrative Biology of the Cell (I2BC), CEA, CNRS, Univ. Paris-Sud, Université Paris-Saclay, 91198 Gif-sur-Yvette cedex, France. [5] LBCMCP, Centre de Biologie Intégrative (CBI), Université de Toulouse, CNRS/UPS, Toulouse, France. [6] The Salk Institute for Biological Studies, Molecular and Cell Biology Department, La Jolla, CA 92037, USA. [7] Department of Gene Mechanisms, Graduate School of Biostudies/The Hakubi Center for Advanced Research, Kyoto University, Yoshida-Konoe-cho, Sakyo-ku, Kyoto 606-8501, Japan. Correspondence and requests for materials should be addressed to A.J.C. (email: tcesare@cmri.org.au)

Inducing genotoxic stress is a common mechanism of action for frontline chemotherapeutics. Overwhelming a cell's capacity to cope with exogenous DNA damage results in lethality through activation of regulated cell death pathways such as apoptosis[1]. Cell death in response to genotoxic stress is more pronounced in actively dividing cells, which imparts the clinical efficacy of genotoxic agents on proliferative cancers. Consistent with the link between proliferation and lethality, many genotoxic agents are associated with cell death during mitosis[2]. Pathways of mitotic death fall under the broad classification of mitotic catastrophe[3].

Mitotic catastrophe is a regulated onco-suppressive mechanism that responds to aberrant mitoses by removing damaged cells from the cycling population[4]. This can occur through cell death or permanent growth arrest. Whilst mitotic catastrophe has been described for some time, there is no clear understanding of the pathway(s) regulating mitotic death[3]. Nevertheless, evidence suggests that mitotic catastrophe is a mechanism of cell death during chemotherapeutic intervention, and that developing resistance to mitotic death would impart a selective advantage to neoplastic cells[5].

Many genotoxic agents directly or indirectly induce replication stress[6]. Replication stress is broadly defined as the slowing or inhibition of DNA replication. This includes endogenous stress originating from oncogene expression, and repetitive and/or secondary DNA structures, or exogenous stress from pharmacological agents that induce DNA damage, reduce nucleotide production, or inhibit DNA polymerases[6,7]. Difficulties arising during DNA replication often manifest as mitotic abnormalities, including chromosome segregation errors, anaphase- or ultra-fine bridges, micronuclei, and the passage of replication stress-induced DNA damage through mitosis[8–11]. While replication stress is implicated as a driver of mitotic catastrophe, mechanisms of replication stress-induced cell death remain unclear.

Here we investigated the cellular response of human primary and cancer cells to pharmacologically induced DNA replication stress with a specific focus on cell lethality. We observed that high dosages of replication stress inducing drugs resulted in mitotic bypass and proliferative arrest without lethality in p53-competent primary fibroblasts. Conversely, in p53-compromised fibroblasts and cancer cells, lethal replication stress induces a striking outcome of spindle assembly checkpoint (SAC)-dependent mitotic arrest and cell death principally conferred during mitosis. We found this mitotic death resulted from two independent pathways engaged during mitotic arrest. A primary pathway regulated by wings apart-like protein homologue (WAPL) that drove cohesion fatigue and BAX/BAK-mediated intrinsic apoptosis, and a secondary pathway signalled through non-canonical telomere deprotection. Together, these data reveal functions for WAPL and telomeres in sensing replication-stress induced damage and executing mitotic death, with implications for cancer therapy and cancer genome evolution.

## Results

**Replication stress induces SAC-dependent mitotic arrest.** To visualize replication stress-induced lethality, we treated human cells with escalating doses of Aphidicolin (APH) or Hydroxyurea (HU) and visualized the outcomes with live-cell imaging (Supplementary Movie 1). APH inhibits family B DNA polymerases, whereas HU inhibits ribonucleotide reductase to limit the nucleoside pool available for nascent DNA synthesis. Cultures were visualized after drug treatment with differential interference contrast (DIC) microscopy every 6 min for up to 65 h (Fig. 1a). Qualitative results indicated a strong mitotic arrest and mitotic death phenotype. We therefore measured mitotic duration from

nuclear envelope breakdown to mitotic exit or mitotic death, and classified mitotic outcomes as normal, death, multipolar cell division (mitosis resulting in more than two daughter cells), or mitotic slippage (transition from mitosis to interphase without cell division) (Fig. 1b).

In primary IMR90 fibroblasts, APH or HU treatment induced a marked reduction in mitotic entry and proliferative arrest (Fig. 1c–e and Supplementary Fig. 1a). The p53 pathway confers growth arrest in response to genomic insult[12], suggesting p53 may prevent mitotic entry in replication stressed cells. In agreement, inhibiting p53 and Rb in IMR90 cells through exogenous expression of HPV16 E6 and E7 (IMR90 E6E7) resulted in a significant increase in mitotic duration and mitotic death with escalating dosages of APH and HU (Fig. 1c, d). Two-dimensional data representation revealed that mitoses which died arose after 20 h of APH or HU treatment, and that mitotic death correlated with increased mitotic duration (Fig. 1f and Supplementary Fig. 1a). IMR90-hTERT cultures expressing the dominant negative p53DD allele also depicted similar mitotic arrest and mitotic death with delayed kinetics in response to lethal APH (Supplementary Fig. 1b, c). We also transduced IMR90 cultures with lentivectors expressing Cas9 and gRNAs targeting p53 and sorted for transduced cells. Analysis of p53 CRISPR targeted populations revealed reduced p53 protein levels and corresponding increases in mitotic duration and mitotic death with APH treatment (Supplementary Fig. 1d–g). Inhibiting p53 is therefore required for replication stress-induced mitotic death in IMR90 cells.

p53-compromised cancer cells also exhibited mitotic arrest and mitotic death with lethal replication stress. HT1080 6TG are a p53 mutant derivative of the HT1080 fibrosarcoma cell line. Treating HT1080 6TG cultures with escalating concentrations of APH and HU revealed concomitant significant increases in mitotic duration and mitotic death (Fig. 1g, h). Mitotic events resulting in death started 20 h after 1 μM APH, or 30 h after 500 μM HU treatment, and correlated with increased mitotic duration (Fig. 1i and Supplementary Fig. 2a). HeLa cervical carcinoma and p53-null Saos-2 osteosarcoma cells also exhibited increased mitotic duration and mitotic death when treated with lethal dosages of APH (Supplementary Fig. 2b, c).

Correlation between mitotic duration and death suggested that mitotic arrest drives replication stress lethality. The SAC is regulated by MPS1 kinase and arrests mitosis until tension is established across the mitotic spindle[13]. We tested SAC involvement in replication stress-induced mitotic arrest by performing live cell imaging of HT1080 6TG cultures treated with APH or HU, and the MPS1 inhibitor reversine[14]. Reversine suppressed mitotic arrest and death, consistent with mitotic arrest being a key determinant of replication stress lethality (Fig. 1j–l and Supplementary Fig. 2d). Additionally, rescuing mitotic death with reversine conferred an increase in multipolar cell division in APH treated cells and mitotic slippage in HU treated cultures (Fig. 1k).

**Replication stress induces death in the same cell cycle.** Mitotic death in multiple p53-compromised cell lines required twenty or more hours of APH or HU treatment. To determine if replication stress-induced lethality occurred in the same or subsequent cell cycle, we created fluorescent, ubiquitination-based cell cycle indicator (FUCCI) expressing HT1080 6TG cultures[15] (Fig. 2a). HT1080 6TG-FUCCI cells were treated with APH or DMSO and visualized with DIC and fluorescent live cell imaging every 6 min for up to 60 h (Supplementary Movie 2). Cells were scored for G1 and S/G2 duration, respectively, by mCherry-hCdt1(30/120) and mVenus-hGeminin(1/110) stability. Mitotic duration and

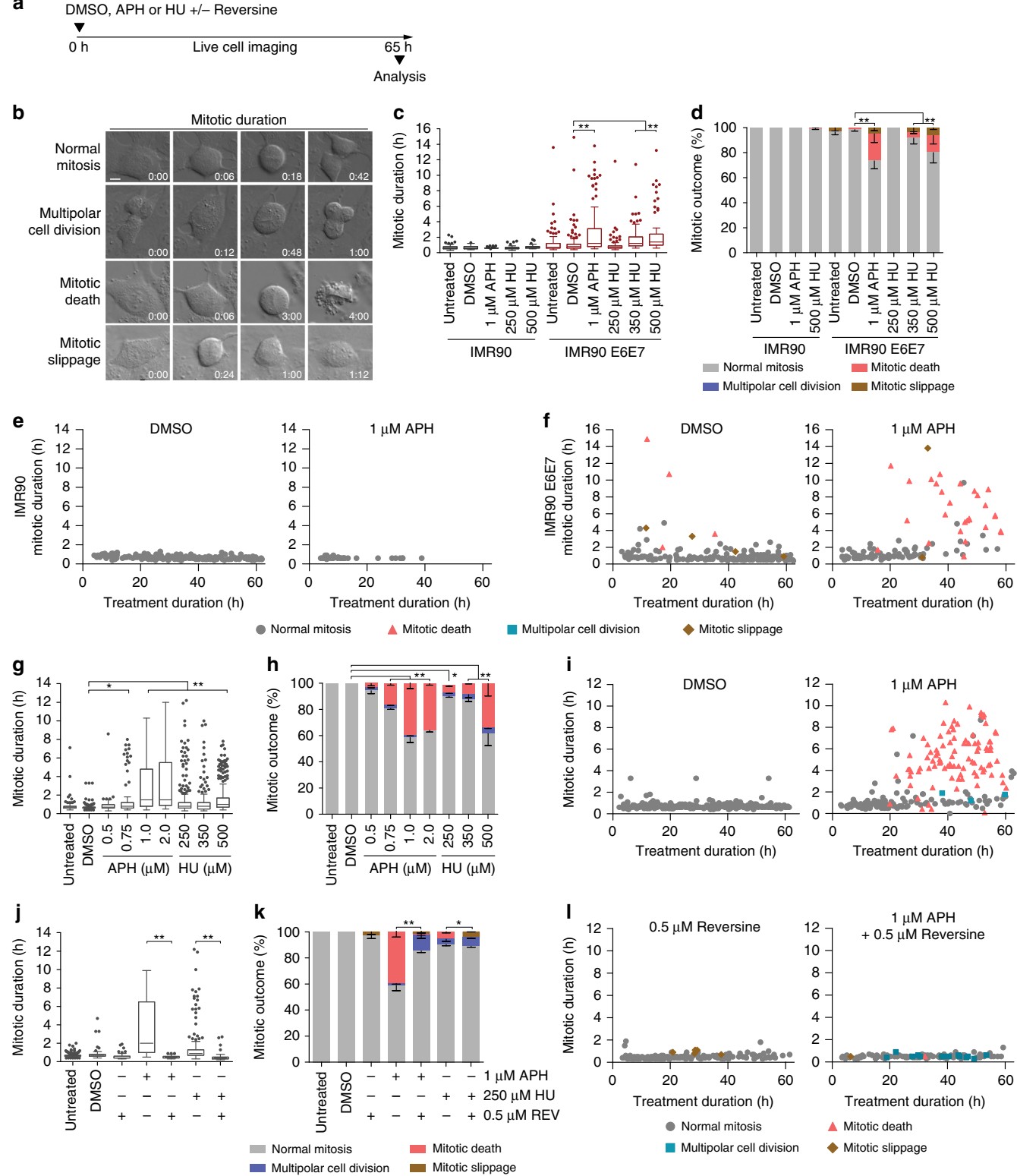

outcomes were classified as described above, with the addition of mitotic bypass, defined as transition from G2 [mVenus-hGemi-nin(1/110) expressing] to G1 [mCherry-hCdt1(30/120) expressing] without mitotic entry (Fig. 2a). We also scored interphase cell death (Fig. 2a).

Asynchronous HT1080 6TG-FUCCI cells displayed a stratified response to APH treatment characterized by cell cycle phase at the time of drug administration. Cells in S/G2 phase when APH was added to the media, and which remained in S/G2 for <20 h with APH, displayed little to no mitotic arrest or cell death phenotype (Fig. 2b, c). Conversely, cells in S/G2 phase when APH was added to the media, and which remained in S/G2 for >20 h with APH, displayed common mitotic arrest and mitotic death in the same cell cycle (Fig. 2b, c). APH did not impact G1 duration (Fig. 2d). However, when cells exited G1 in the presence of APH they displayed a subsequent extended S/G2, mitotic arrest and

**Fig. 1** Replication stress induces mitotic arrest and mitotic death. **a** Time course of live cell imaging throughout this study unless indicated otherwise. **b** Representative images depicting mitotic outcome and duration from DIC live cell imaging experiments. Time is shown as (hours: minutes) relative to the first image of the series. Scale bar represents 10 μm. **c** Mitotic duration of IMR90 or IMR90 E6E7 cells following treatment with Dimethyl sulfoxide (DMSO) vehicle, Aphidicolin (APH), or Hydroxyurea (HU) (three biological replicates scoring $n \geq 46$ mitoses per condition for IMR90 or $n \geq 66$ mitoses per condition for IMR90 E6E7 are compiled in a Tukey box plot, Mann–Whitney test). **d** Outcome of the mitotic events in (**c**) (mean ± s.e.m, $n = 3$ biological replicates, Fisher's Exact Test). **e, f** Two-dimensional dot plots for IMR90 (**e**) and IMR90 E6E7 (**f**) of mitotic duration and outcome for the data shown in (**c, d**). Each symbol represents an individual mitosis. $T = 0$ h is when DMSO or APH were added to the culture. Location of a symbol on the x-axis indicates the time after treatment when that mitosis initiated, the height on the y-axis represents mitotic duration, and the symbol corresponds to mitotic outcome. **g** Mitotic duration of HT1080 6TG cells following treatment with DMSO, APH, or HU (three biological replicates scoring $n \geq 92$ mitotic events per condition compiled in a Tukey box plot, Mann–Whitney test). **h** Outcome of the mitotic events in (**g**) (mean ± s.e.m, $n = 3$ biological replicates, Fisher's Exact Test). **i** Two-dimensional dot plots of mitotic duration and outcome from (**g, h**). **j** Mitotic duration of HT1080 6TG cultures treated with DMSO, APH or HU ± reversine (REV) (three biological replicates scoring $\geq 112$ mitotic events per each condition compiled in a Tukey box plot, Mann–Whitney test). **k** Outcome of mitotic events in (**j**) (mean ± s.e.m., $n = 3$ biological replicates, Fisher's Exact Test). **l** Two-dimensional plots of mitotic duration and outcome from (**j, k**). For all panels, *$p < 0.05$, **$p < 0.01$. Source data are provided as a Source Data file

mitotic death all in the same cell cycle (Fig. 2b, d). Notably, 85% of cell death events in HT1080 6TG-FUCCI cells observed in the first cell cycle with APH treatment occurred during mitosis (Fig. 2e). Cumulatively, these data indicate lethal replication stress administered early in S-phase predominantly kills cells in the immediately following mitosis.

To determine if persistent APH treatment during mitotic arrest impacts mitotic death, we imaged HT1080 6TG-FUCCI cells during APH treatment and following washout of the drug from the growth media. Cells were cultured with 1 μM APH for 30 h before media removal and replacement with culture media containing DMSO or fresh 1 μM APH (Supplementary Fig. 3a). Notably, APH washout failed to rescue lethality in cells already arrested in mitosis (Supplementary Fig. 3b). Additionally, for 2 h after APH removal, cells continued to enter mitosis, arrest and die (Supplementary Fig. 3b). Mitotic arrest and death thus did not result from persistent APH treatment during cell division. Indicating instead that an outcome induced by lethal replication stress, prior to mitotic entry, confers the subsequent arrest and mitotic death.

**Replication stress induces p53-dependent mitotic bypass.** To determine how replication stress impacted cell cycle progression in p53-competent primary fibroblasts, we live-cell imaged IMR90 FUCCI cultures. APH treatment of IMR90-FUCCI cells induced an extended S/G2, mitotic bypass, and growth arrest in presumably tetraploid G1-phase cells (Supplementary Fig. 3c). This is consistent with p53-activation from a genomic DNA damage response (DDR) being sufficient to induce mitotic bypass and proliferative arrest[16,17]. We note APH treated IMR90 FUCCI cells often displayed what appeared as a sustained G1-arrest without S-phase entry (Supplementary Fig. 3c), likely reflecting early S-phase arrest when mCherry-hCdt1(30/120) remains stable. p53 thus prevents mitotic entry, and therefore mitotic death, in replication stress stressed IMR90 cultures. For the remainder of this study we focused on mitotic death in p53-compromised cells.

**Replication stress induces distinct types of mitotic death.** The specific contribution of cell death pathways in mitotic catastrophe remains poorly understood[3]. BAX and BAK are BCL-2 family proteins essential for apoptosis through their role in mitochondrial outer membrane permeabilization (MOMP)[18,19]. Using CRISPR-Cas9 we generated clonal HeLa and HT1080 6TG *BAX* and *BAK* double knock out (*BAX BAK* DKO) cell lines (Supplementary Fig. 4a). Parental and *BAX BAK* DKO cells were treated with APH and visualized with live cell imaging. APH induced mitotic arrest in *BAX BAK* DKO cultures, with individual mitotic events exhibiting a longer duration mitotic arrest

than observed in parental cells (Fig. 3a, b and Supplementary Fig. 4b–d). Of note, *BAX BAK* DKO rescued most, but not all, mitotic death in APH treated cultures at the cost of increased multipolar cell division and mitotic slippage (Fig. 3c and Supplementary Fig. 4e).

Observation of apoptosis-dependent and -independent mitotic death suggested the potential for multiple mitotic death initiating events occurring simultaneously in response to replication stress. To assess this possibility, we imaged live cultures of H2B-mCherry expressing HT1080 6TG cultures treated with a lethal dose of APH. These experiments revealed two chromosome phenotypes associated with replication stress-induced mitotic death (Fig. 3d and Supplementary Movie 3). Type 1 mitotic death was defined by a dispersion of chromosomal material immediately preceding lethality, whereas Type 2 mitotic death was defined by the collapse or condensation of the genetic material without chromosome dispersion (Fig. 3d). Type 1 mitotic death was the dominant outcome in both 0.75 and 1.0 μM APH treated cultures, followed by Type 2 mitotic death, with a minor percentage of interphase lethality occurring exclusively in the 1.0 μM APH treated cells (Fig. 3e).

**Replication stress induces WAPL-dependent cohesion fatigue.** We characterized the chromosome phenotypes associated with mitotic death by preparing cytogenetic chromosome preparations from APH or HU treated HT1080 6TG cultures (Fig. 4a, b and Supplementary Fig. 5a, b). This revealed a major phenotype of aberrant chromosome cohesion induced by replication stress (Fig. 4b, c and Supplementary Fig. 5b). We sub-classified the cohesion abnormalities as a mild phenotype defined by visible separation of sister centromeres; a moderate phenotype of cohesion loss between sister centromeres and the adjacent chromosomal region, with cohesion remaining at distal chromosome arms; and a severe phenotype of complete sister chromatid separation (Fig. 4b). Replication stress also induced entangled chromosomes consistent with condensation defects (Supplementary Fig. 5a)[20].

The mild and moderate cohesion phenotypes initiated from the centromeres (Fig. 4b), suggesting involvement of microtubule pulling forces driving chromatid separation. We tested this by treating cells with a lethal dose of APH or HU in combination with colcemid or Taxol to inhibit microtubule dynamics, or reversine to silence the SAC (Fig. 4a). In APH and HU treated cells, Taxol, colcemid, and reversine all suppressed the moderate and severe cohesion phenotype (Fig. 4c and Supplementary Fig. 5b). These observations are consistent with cohesion fatigue, a phenomena where sustained microtubule pulling forces exerted during mitotic arrest drive unscheduled chromosome segregation prior to anaphase onset[21,22]. We interpret these data to indicate

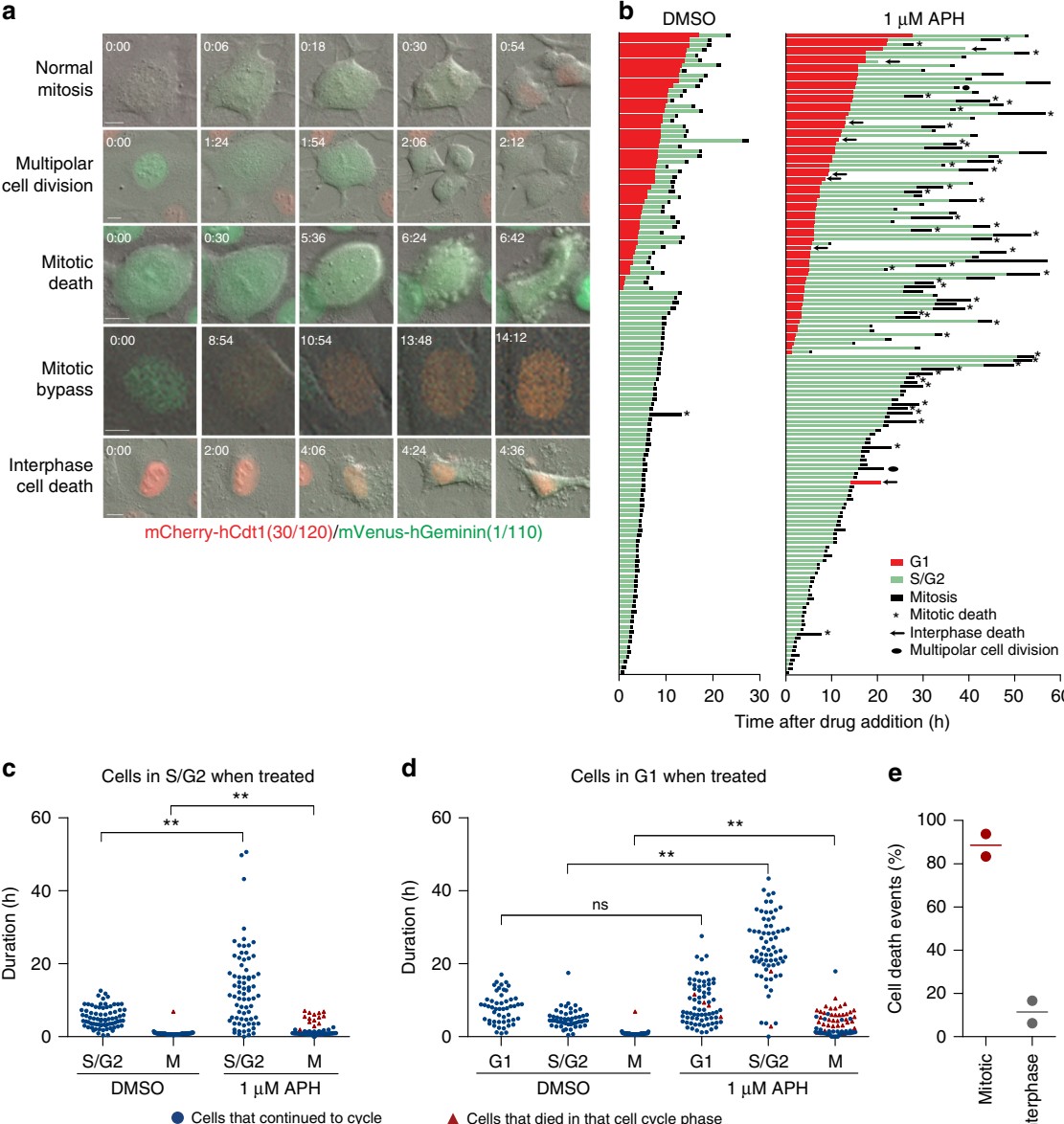

**Fig. 2** Replication stress induces mitotic death in the same cell cycle. **a** Representative images from live cell microscopy of HT1080 6TG-FUCCI cells. Time is shown as (h:min) relative to the first image of the series. Scale bars represent 10 µm. **b** Cell fate map of HT1080 6TG-FUCCI live cell imaging. Each bar represents an individual cell as it progresses through the first cell cycle to cell division or death, relative to addition of DMSO ($n = 127$) or APH ($n = 148$) to the growth media at $T = 0$. Segment length represents the duration a cell spent in each cell cycle phase. Cell cycle phases are colour coded according to FUCCI and shown in the legend. Data are from two independent biological replicates compiled into a single graph. **c**, **d** Cell cycle phase duration and outcome for HT1080 6TG-FUCCI live cell imaging shown in (**b**). Data are sorted based on cells that were in S/G2 (**c**) or G1 (**d**) at the time the DMSO or APH was administered. Symbols indicate if a cell survived that cell cycle phase and progressed, or if the cell died during that cell cycle phase. Data are from two independent biological replicates, compiled into a dot plot (Mann–Whitney test, $**p < 0.01$, ns = not significant). **e** Categorization of all cell death events from (**b**) in the first cell cycle with APH treatment ($n = 2$ replicates, line represents the mean). Source data are provided as a Source Data file

the moderate and severe conditions represent a phenotypic continuum of cohesion fatigue as sister chromatids are progressively separated during mitotic arrest. Conversely, the mild cohesion phenotype was not impacted by reversine, Taxol or colcemid, indicating instead that the mild phenotype results from replication stress, but not mitotic arrest nor microtubule-dependent forces. Chromosome entanglement occurred in all replication stress conditions (Fig. 4c and Supplementary Fig. 5a–c). However, the complexity of entangled chromosomes prevented classification of cohesion status and was excluded from subsequent analysis.

WAPL is a cohesion antagonist that regulates opening of the cohesion DNA exit gate to enable dynamic interaction between cohesion and chromatin[23,24]. Depleting WAPL increases cohesin retention on chromatin and suppresses cohesion fatigue[21,24,25]. To probe for WAPL involvement in the replication stress-induced cohesion phenotypes, we siRNA depleted WAPL, treated cultures with a lethal dose of APH or HU, and assayed chromosome cohesion in cytogenetic chromosome preparations (Fig. 4d–g and Supplementary Fig. 5c). We observed in APH and HU treated HT1080 6TG cultures that WAPL depletion suppressed both the moderate and severe sister chromatid cohesion phenotypes

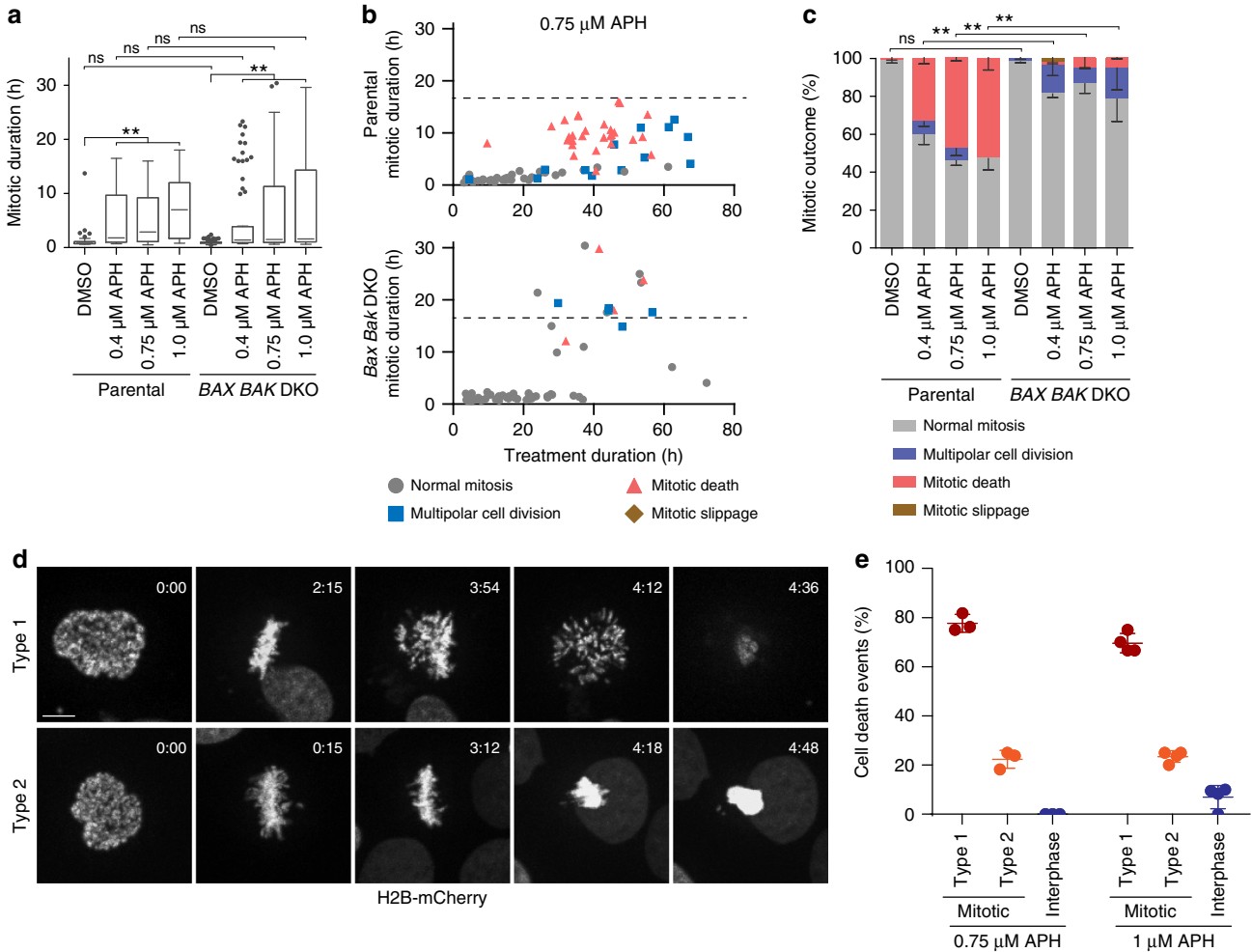

**Fig. 3** Replication stress induces distinct types of mitotic death. **a** Mitotic duration of HeLa parental and *BAX BAK* DKO cells following treatment with DMSO or APH (three biological replicates using independent clones scoring $n \geq 32$ mitotic events per condition are compiled in a Tukey box plot, Mann–Whitney test, **$p < 0.01$, ns = not significant). **b** Two-dimensional dot plots of mitotic duration and outcome in 0.75 μM APH treated HeLa parental and *BAX BAK* DKO cells from (**a**). The dashed line identifies the longest duration mitosis observed in the parental cells. **c** Outcome of the mitotic events in (**a**) (mean ± s.e.m., $n = 3$ biological replicates using independent clones, Fisher's Exact Test, **$p < 0.01$, ns = not significant). **d** Representative images of Type 1 and Type 2 mitotic death in 1 μM APH treated HT1080 6TG H2B-mCherry cells. Time is shown as (h:min) relative to the first image of the series. Scale bar represents 10 μm. **e** Quantitation of cell death events in 0.75 and 1.0 μM APH treated HT1080 6TG H2B-mCherry cultures (mean ± s.d., $n = 3$ biological replicates scoring ≥57 mitotic events per condition for 0.75 μM APH and $n = 4$ biological replicates scoring ≥91 mitotic events per condition for 1.0 μM APH). Source data are provided as a Source Data file

dependent upon mitotic arrest, and the mitotic arrest-independent minor cohesion phenotype (Fig. 4f–g and Supplementary Fig. 5c).

**Type 1 mitotic death is WAPL and BAX/BAK dependent**. We also assayed the impact of WAPL depletion on replication stress-induced mitotic lethality. Concomitant with rescue of the cohesion phenotypes, WAPL depletion significantly reduced mitotic duration and rescued mitotic death in HT1080 6TG cultures treated with 1 μM APH or 350 μM HU (Fig. 5a, b and Supplementary Fig. 6a, b). This was not an artefact of reduced proliferation, as APH and HU treated cells continued to enter mitosis in WAPL depleted cultures (Supplementary Fig. 6c). Similarly, WAPL depletion in 1 μM APH treated HeLa cultures significantly reduced mitotic duration and mitotic death at the cost of a significant increase in multipolar cell divisions (Supplementary Fig. 6d, e).

As both WAPL depletion and *BAX BAK* DKO rescued most of the mitotic death induced by replication stress, we reasoned that

WAPL and BAX/BAK may function in the same pathway. To test this, we created HT1080 6TG H2B-mCherry *BAX BAK* DKO cells (Supplementary Fig. 6f). We then treated HT1080 6TG H2B-mCherry, WAPL depleted HT1080 6TG H2B-mCherry, or HT1080 6TG H2B-mCherry *BAX BAK* DKO cells with 1 μM APH and visualized mitotic outcomes with live cell imaging (Fig. 5c, d and Supplementary Fig. 6g). WAPL depletion completely suppressed Type 1 mitotic death with no significant effect on Type 2 mitotic or interphase death (Fig. 5d). Similarly, *BAX BAK* DKO also largely suppressed Type 1 mitotic death with no significant effect on Type 2 mitotic or interphase death (Fig. 5d). Together, these data indicate that Type 1 mitotic death is WAPL and BAX/BAK dependent.

**Rescuing mitotic death results in distinct outcomes**. Additionally, we quantified how suppressing mitotic death in HT1080 6TG H2B-mCherry cells impacted cellular outcomes. In the absence of replication stress, WAPL depletion induced a slight but significant increase in chromosome segregation errors

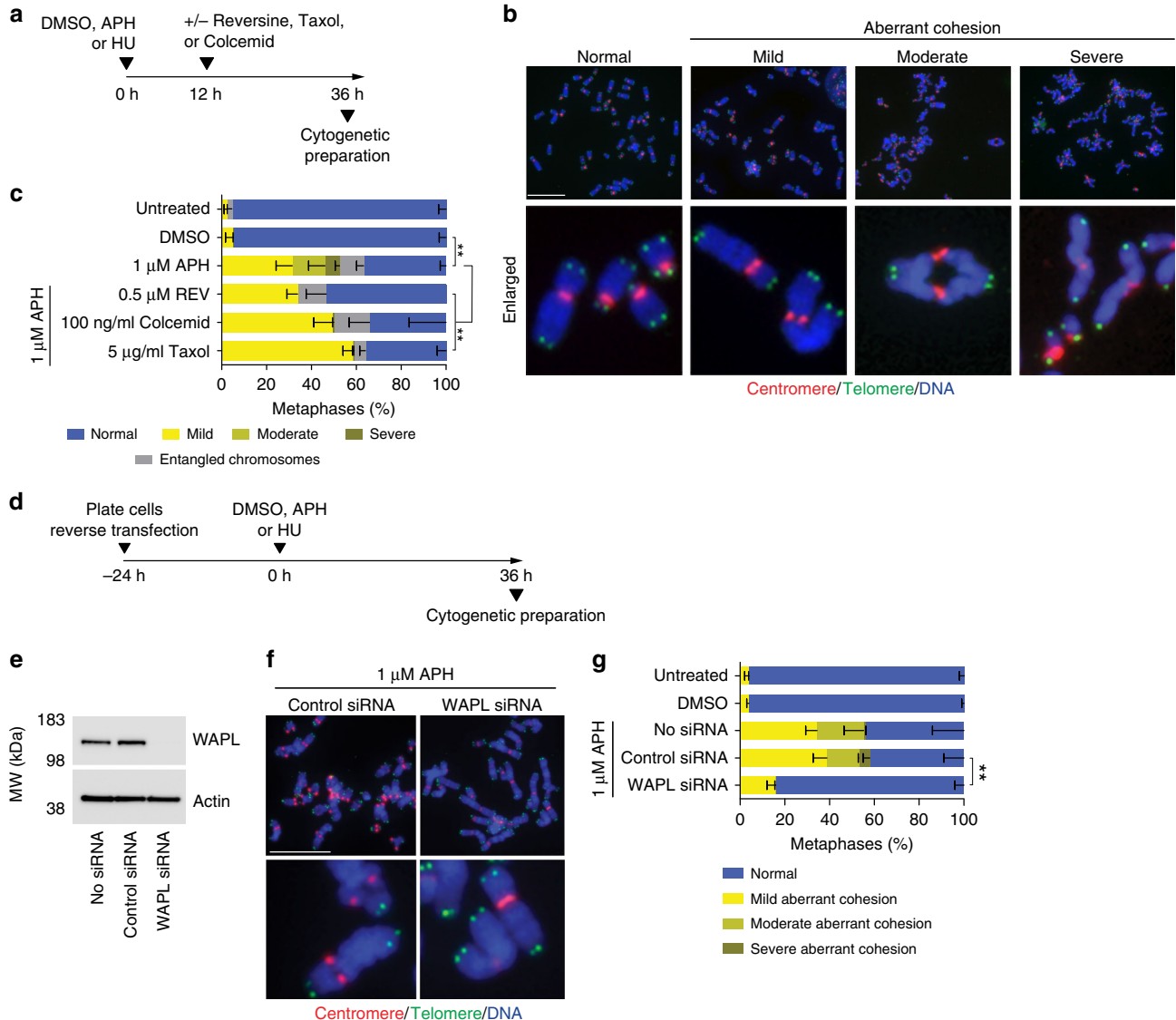

**Fig. 4** Replication stress induces WAPL-dependent cohesion fatigue. **a** Time course for experimentation in (**b**, **c**). **b** Representative images of cytogenetic chromosome spreads depicting the replication stress-induced cohesion phenotypes from HT1080 6TG cells stained with DAPI (blue), centromere (red) and telomere (green) FISH. **c** Quantitation of cohesion phenotypes depicted in (**b**) in HT1080 6TG cells treated with DMSO or APH, ± reversine (REV), colcemid or Taxol (mean ± s.e.m., $n = 3$ biological replicates scoring ≥ 152 chromosome spreads per condition, Fisher's Exact Test). **d** Experimental time course for (**e–g**). **e** Western blots of whole cell extracts from HT1080 6TG cells treated with WAPL or control siRNA. **f** Representative images of cytogenetic chromosome spreads from HT1080 6TG cells treated with APH and siRNA. **g** Quantitation of cohesion phenotypes in HT1080 6TG cells treated with APH ± siRNA (mean ± s.e.m., $n = 3$ biological replicates scoring ≥ 144 chromosome spreads per condition, Fisher's Exact Test). For all panels **p < 0.01. Scale bars represent 10 μm. Source data are provided as a Source Data file

(6.0% ± 2.4 of mitoses, Supplementary Fig. 6g). However, when combined with 1 μM APH, WAPL depletion conferred a substantial increase in multipolar cell division or mitoses resulting in micronuclei formation (47.2% ± 2.8 of mitoses, Fig. 5d)[26]. Alternatively, APH treatment in HT1080 6TG H2B-mCherry *BAX BAK* DKO cells induced a significant increase in mitotic slippage (Fig. 5d).

An explanation for these distinct outcomes is that WAPL and BAX/BAK function sequentially in the Type 1 mitotic death pathway. Data presented above suggests that WAPL functions upstream to regulate mitotic arrest and cohesion fatigue, whereas the known BAX/BAK function in MOMP[19] is consistent with BAX/BAK functioning downstream in apoptotic induction. Depleting WAPL in *BAX BAK* DKO cells suppressed mitotic duration in APH treated cultures, consistent with WAPL

promoting upstream mitotic arrest (Fig. 5e, f). Assaying cohesion in APH treated *BAX BAK* DKO cells revealed a specific increase in mitoses with the severe cohesion phenotype (Fig. 5g), consistent with a downstream inability to clear mitoses following complete cohesion fatigue. Close examination using live-cell confocal microscopy of 1 μM APH treated H2B-mCherry *BAX BAK* DKO cells revealed a striking outcome of multi-lobular nuclei accompanying mitotic slippage (Supplementary Movie 4). We expect this occurs as result of the improper nuclear packaging of fully separated sister chromatids. Remnants of chromosome segregation errors persisted after 1 μM APH was applied to cultures for 2 days, removed, and cells allowed to propagate for one additional day, with multilobular nuclei significantly more enriched in *BAX BAK* DKO cells (Fig. 5h–j).

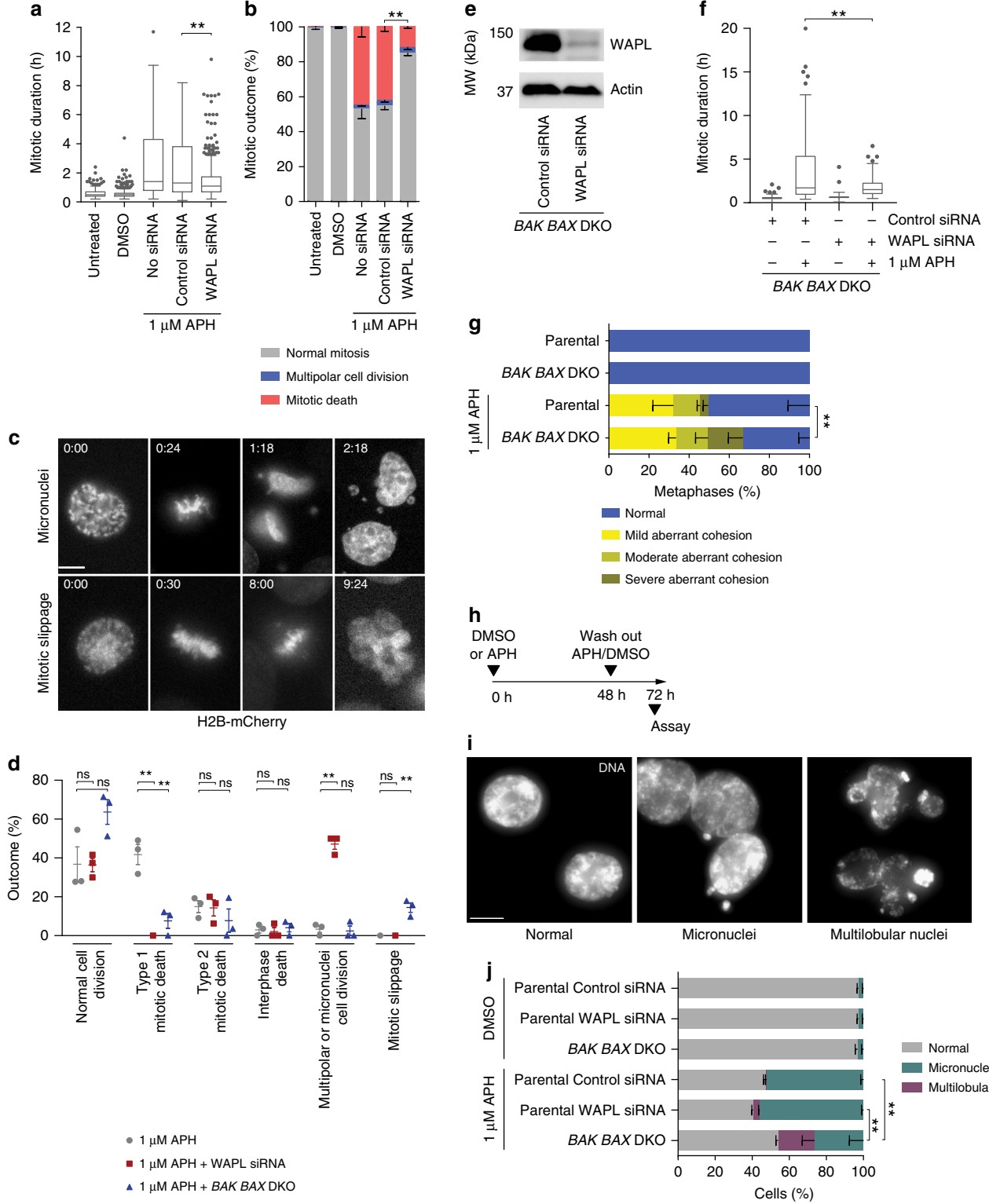

**Replication stress induces mitotic telomere deprotection**. To address potential mechanisms contributing to the minor Type 2 mitotic death pathway, we assayed DDR activation on cytocentrifuged mitotic chromatin following treatment with lethal dosages of APH or HU (Fig. 6a, b). Surprisingly, we observed similar numbers of genomic and telomeric γ-H2AX foci on mitotic chromosomes from APH and HU treated cells (Fig. 6c). Telomeres represent a minor proportion of the total genome, suggesting induction of a telomere-specific phenotype with lethal

replication stress. One possibility is that exogenous replication stress disproportionally impacts the difficult to replicate repetitive G-rich telomeric DNA sequence[27–29]. Replication stress in vertebrate telomeres is attenuated through the telomere-specific TRF1 protein[29].

To investigate potential contribution of telomere replication stress to mitotic lethality, we over-expressed mCherry-TRF1 (mCherry-TRF1$^{OE}$) in HT1080 6TG cells (Fig. 6d). Surprisingly, mCherry-TRF1$^{OE}$ failed to reduce the number of mitotic

**Fig. 5** Type 1 mitotic death is WAPL and BAX/BAK dependent. **a** Mitotic duration of HT1080 6TG cells treated with APH ± siRNA (three biological replicates scoring $n \geq 429$ mitoses per condition are compiled in a Tukey box plot, Mann–Whitney test). **b**) Outcome of the mitotic events in (**a**) (mean ± s.e.m., $n = 3$ biological replicates, Fisher's Exact Test). **c** Representative images of cell division generating micronuclei, or mitotic slippage, in APH treated HT1080 6TG H2B-mCherry cells. Time is shown as (h:min) relative to the first image in the series. **d** Mitotic outcomes in HT1080 6TG H2B-mCherry cells treated with APH ± WAPL siRNA or *BAX BAK* DKO (three biological replicates scoring $n = 97$ APH, $n = 38$ APH + WAPL knock down, $n = 88$ APH + *BAX BAK* DKO cells, Student t-test or Fisher's Exact test). **e** Western blots of whole cell extracts from HT1080 6TG *BAX BAK* DKO cells treated with WAPL or control siRNA. **f** Mitotic duration of HT1080 6TG *BAX BAK* DKO cells treated with APH ± siRNA (three biological replicates scoring $n \geq 86$ mitoses per condition are compiled in a Tukey box plot, Mann–Whitney test). **g** Quantitation of cohesion phenotypes as depicted in Fig. 4b in HT1080 6TG *BAX BAK* DKO cells treated with DMSO or APH (mean ± s.e.m, $n = 3$ biological replicates scoring $\geq 50$ chromosome spreads per condition, Fisher's Exact Test). **h** Timeline of the experiment shown in (**i, j**). **i** Images of DAPI stained nuclei depicting the indicated nuclear phenotypes. **j** Quantitation of the phenotypes shown in (**i**) from HT1080 6TG cells and derivatives at the 72-hour time point as indicated in (**h**) ($n = 3$ biological replicates scoring 100 nuclei per replicate, mean ± s.e.m, Fisher's exact test). For all panels ns = not significant, **$p < 0.01$. Scale bars represent 10 μm. Source data are provided as a Source Data file

telomere DDR foci observed with APH treatment (Fig. 6e, f). mCherry-TRF1[OE] also failed to impact mitotic duration or confer a survival advantage to cells treated with 1 μM APH (Fig. 6g, h). Together this indicates that the mitotic outcomes observed with lethal APH treatment are independent of replication stress within the telomeric DNA.

Alternatively, the mitotic telomere DDR accompanying lethal replication stress could arise through non-canonical telomere deprotection (Supplementary Fig. 7a)[30]. Canonical telomere function is to mediate chromosome end protection and regulate cell aging as a function of telomere length[31]. However, telomere-specific DDR activation also occurs through a non-canonical pathway during prolonged mitotic arrest[30]. Mitotic arrest-dependent telomere deprotection is regulated by the Aurora B kinase and the telomere-specific TRF2 protein[30]. During mitotic arrest in human cells, TRF2 dissociates from chromosome ends resulting in a telomere macromolecular structural change from telomere-loops (t-loops) to linear telomeres[30,32]. This exposes the chromosome end as an ATM substrate to activate the mitotic telomere DDR[32].

To test for mitotic arrest-dependent telomere deprotection we treated HT1080 6TG cells with lethal dosages of APH or HU in combination with the SAC inhibitor reversine, the Aurora B inhibitor Hesperadin, or the ATM inhibitor KU-55933 (Fig. 6i, j)[33,34]. Consistent with non-canonical mitotic telomere deprotection, the telomere DDR induced by lethal replication stress was suppressed by inhibiting Aurora B, ATM, or mitotic arrest (Fig. 6j and Supplementary Fig. 7b). Reversine and Hesperadin did not reduce genomic DDR foci induced by APH or HU, indicating the reduced number of telomeric γ-H2AX foci did not result from global DDR suppression (Fig. 6i, j and Supplementary Fig. 7b). KU-55933 did reduce genomic DDR foci induced by APH or HU, consistent with the global role for ATM in DDR signalling. Additionally, no telomere shortening occurred with the telomere DDR in 1 μM APH treated cells, (Fig. 6k), consistent with telomere length independent non-canonical deprotection[30]. We note that mitotic telomere deprotection occurred in parallel with cohesion fatigue as we observed DDR-positive telomeres in mitoses with completely separated sister chromatids (Supplementary Fig. 7c).

**Telomere deprotection promotes replication stress lethality.** Mitotic telomere deprotection is regulated by TRF2[30]. While *TRF2* deletion results in the complete loss of telomere protection and end-to-end chromosome fusions[35], partial TRF2 depletion induces intermediate-state telomeres that activate ATM but suppress covalent telomere ligation[36,37]. TRF2 shRNAs of differing efficiency induce only intermediate-state telomeres (TRF2 sh-F), or both intermediate-state telomeres and end-to-

end chromosome fusions (TRF2 sh-G)[36] in HT1080 6TG cells. TRF2 depletion and over-expression (TRF2[OE]) also, respectively, sensitize and suppress non-canonical mitotic telomere deprotection[30].

To determine if mitotic telomere deprotection impacts replication stress-induced mitotic death, we created HT1080 6TG TRF2 sh-F and HT1080 6TG TRF2[OE] cell lines (Fig. 7a). Cell line characterization identified that TRF2[OE] and TRF2 sh-F did not impact genomic DNA replication rates, or cohesion fatigue induced by replication stress (Supplementary Fig. 8a-c). Additionally, no telomere fusions occurred with TRF2 sh-F as a potential source of mitotic arrest[38] (Supplementary Fig. 8d–f). While TRF2 was implicated in peri-centromeric DNA replication[39], we observed no effect of TRF2 sh-F or TRF2[OE] on centromeric γ-H2AX foci in APH treated mitotic cells (Supplementary Fig. 8g). These data indicate that the mitotic phenotypes described below which associate with altered TRF2 expression result from telomere-specific outcomes.

Consistent with previous observations, TRF2 sh-F induced a mitotic telomere-specific DDR in the absence of replication stress[36]. However, the telomere DDR in TRF2 sh-F cells was significantly amplified in 1 μM APH treated cells (Fig. 7c, e). Conversely, TRF2[OE] suppressed mitotic telomere deprotection in response to replication stress (Fig. 7d, e). TRF2 sh-F and TRF2[OE] had an insignificant or very modest impact on mitotic duration with 1 μM APH (Supplementary Fig. 8h). However, TRF2 sh-F significantly reduced the duration of mitotic arrest until cell death in 1 μM APH treated cultures (Fig. 7f). Additionally, TRF2 sh-F induced a significant increase in mitotic death in 1 μM APH treated cells, while TRF2[OE] conferred a significant suppression of mitotic death under the same conditions (Fig. 7g). Cumulatively the data are consistent with mitotic telomere deprotection contributing to a minor proportion of mitotic death in response to lethal replication stress.

## Discussion
Here we identify that cell death induced by replication stress occurs predominantly in mitosis through parallel pathways regulated by WAPL and telomere deprotection (Fig. 8). Further, we find that suppressing mitotic death promotes distinct cellular outcomes through diverse mechanisms depending on how the cell death is averted. As replication stress is a principle driver of genome instability in oncogenesis[9], and many frontline chemotherapeutics directly or indirectly induce replication stress[6], our findings provide insight into mechanisms of cell death, drug resistance, and genome evolution during cancer therapy.

Using extended duration live-cell imaging we identified that lethal replication stress induces SAC-dependent mitotic arrest and mitotic death in diverse p53-compromised human cell lines. In asynchronous cultures, cells in G1 at the time of APH

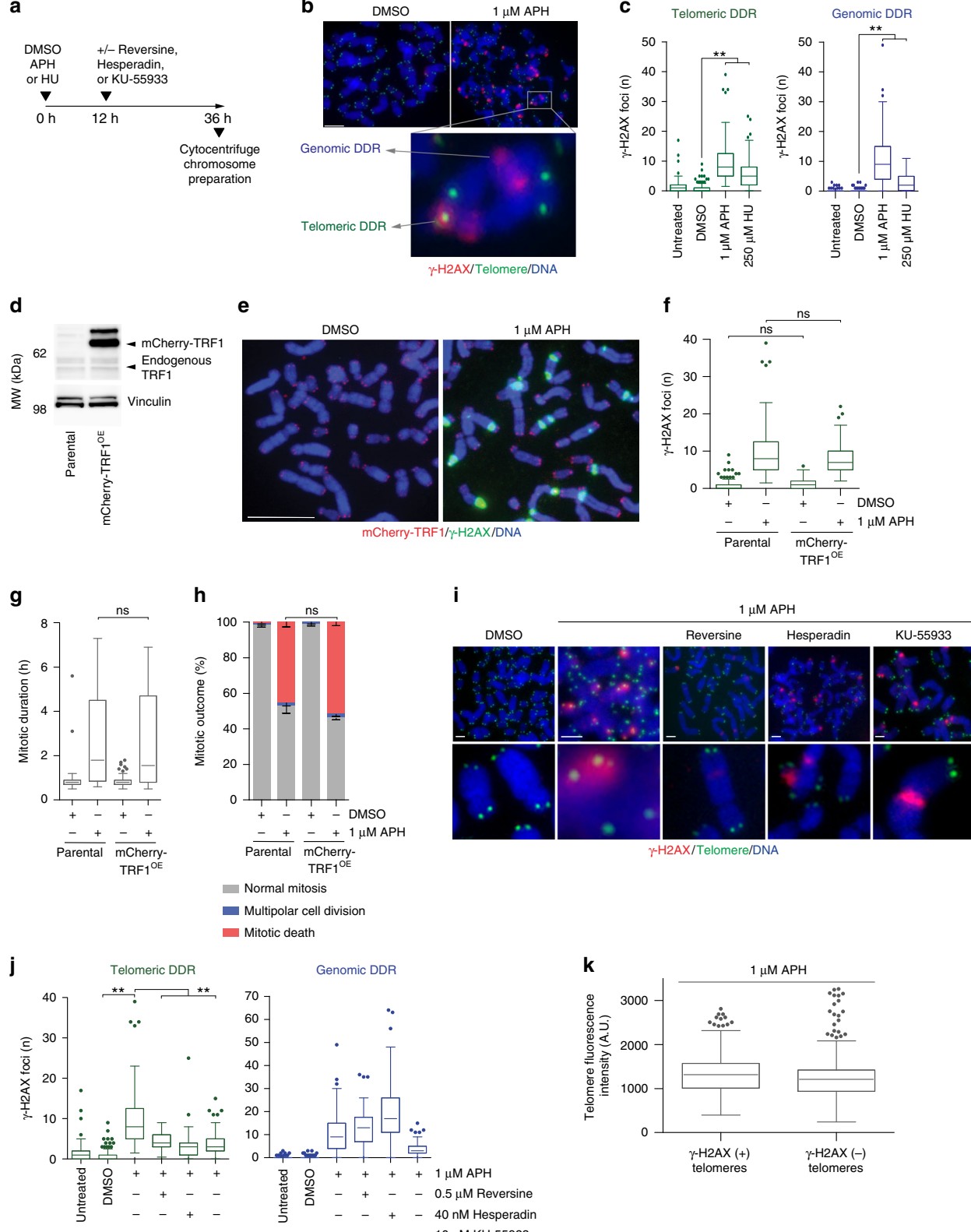

administration, or cells in S/G2 that took greater than 20 h with APH to enter mitosis, had the highest incidence of mitotic death. We interpret this result to indicate that replication stress encountered in early S-phase conferred the greatest probability for subsequent mitotic lethality. Notably, lethal replication stress drove mitotic death in the same cell cycle, suggesting mitotic death is an active process to immediately eliminate compromised cells in the following mitosis. We anticipate this prevents accumulation of genetically unstable tissue by preventing proliferation of cells following excessive replication stress.

Replication stress is repaired during mitosis through a mechanism dependent POLD3, a component of DNA polymerase δ which is inhibited by APH[40–42]. However, APH washout failed to rescue the arrest or lethality in cells already in mitosis, or cells

**Fig. 6** Replication stress induces mitotic telomere deprotection. **a** Experimental time course for (**b**, **c**). **b** Example images of cytocentrifuged chromosome spreads from HT1080 6TG cells treated with DMSO or APH and stained with DAPI (blue), γ-H2AX immunofluorescence (red) and telomere FISH (green). **c** Quantitation of mitotic telomeric and genomic DDR foci in APH or HU treated HT1080 6TG cells (three biological replicates scoring $n \geq 81$ chromosome spreads per condition are compiled in a Tukey box plot, Mann–Whitney test). **d** Western blot of whole cell extracts from HT1080 6TG parental and HT1080 6TG mCherry-TRF1$^{OE}$ cells. **e** Representative images of cytocentrifuged chromosomes from HT1080 6TG mCherry-TRF1$^{OE}$ cells treated with DMSO or APH, stained with DAPI (blue) and γ-H2AX IF (green). **f** Quantitation of mitotic telomeric DDR foci in HT1080 6TG parental and HT1080 6TG mCherry-TRF1$^{OE}$ cells (two biological replicates scoring $n \geq 48$ chromosome spreads per condition are compiled in a Tukey box plot, Mann–Whitney test). **g** Mitotic duration of HT1080 6TG parental and HT1080 6TG mCherry-TRF1$^{OE}$ cells with DMSO or APH treatment (three biological replicates scoring $n \geq 57$ mitoses per condition compiled in a Tukey box plot, Mann–Whitney test). **h** Outcome of the mitotic events in (**g**) (mean ± s.e.m., $n = 3$ biological replicates, Fisher's Exact Test). **i** Representative images of the mitotic DDR on cytocentrifuged chromosome as shown in (**b**) for HT1080 6TG cells treated with APH ± reversine, Hesperadin or KU-55933. **j** Quantitation of mitotic telomere and genomic DDR foci for the conditions shown in (**i**) (three biological replicates scoring $n \geq 37$ chromosome spreads per condition are compiled in a Tukey box plot, Mann–Whitney test). **k** Telomere PNA FISH signal intensity in arbitrary units (A.U.) from individual telomeres in APH treated HT1080 6TG cells, sorted by γ-H2AX status (three biological replicates scoring $n \geq 1529$ telomeres per condition compiled into a Tukey box plot, Mann–Whitney test). For all panels, ns = not significant, **$p < 0.01$. Scale bars represent 10 μm. Source data are provided as a Source Data file

entering mitosis in the following 2 h. Mitotic arrest is therefore not a result of inhibiting mitotic DNA synthesis. SAC maintenance thus resulted from passage of a phenotype induced by replication stress into mitosis. We suggest this is linked to the minor cohesion phenotype. In contrast to the moderate and severe phenotypes, the mild cohesion phenotype was dependent on replication stress but not mitotic arrest nor microtubule dynamics. Failure to establish or maintain cohesion at sister centromeres is expected to impact spindle tension and maintain the SAC[43,44]. WAPL depletion simultaneously rescued both the minor cohesion phenotype and mitotic arrest in replication stressed cells, consistent with WAPL functioning upstream to promote mitotic arrest in response to lethal replication stress.

Our observation of two types of mitotic death induced by replication stress reconciles conflicting reports on cell death mechanisms during mitotic catastrophe, by indicating that replication stress simultaneously engages multiple cell death pathways. The predominant Type 1 pathway was defined by cohesion fatigue and BAX/BAK-mediated intrinsic apoptosis. Cytogenetically, we observed that chromosomes in a mitotic cell typically displayed similar cohesion phenotypes, suggesting that cohesion fatigue progresses uniformly during replication stress-induced mitotic arrest. Mitoses with completely separated sister chromatids accumulated in 1 μM APH treated *BAX BAK* DKO cells consistent with a downstream failure to induce apoptosis following complete cohesion fatigue. We were unable to identify how cohesion fatigue signals apoptosis. However, one intriguing possibility is that sudden loss of spindle tension induces signalling to activate the apoptotic cascade. Because cohesion fatigue requires microtubule pulling forces, replication stress-induced mitotic death may differ mechanistically from cell death following treatment with mitotic poisons that inhibit microtubule dynamics.

A significant finding of our study is that WAPL plays a major role mediating replication stress-lethality. We anticipate WAPL function extends beyond simply driving cohesion fatigue. Cohesin dynamically interacts with chromatin and stalled replication forks[45], and inhibiting WAPL-dependent Cohesin mobilization impacts cell viability in budding yeast treated with replication stress inducing drugs[46]. WAPL also regulates three-dimensional genome architecture though chromatin looping[47–50], and chromatin looping organizes replication origins to facilitate DNA replication[51]. Consistent with a more expansive role for WAPL in regulating replication stress lethality, we observed that WAPL depletion suppressed both the minor cohesion phenotype and mitotic arrest in APH or HU treated cells. This indicates WAPL regulates centromeric cohesion in response to replication stress to maintain SAC activation in the following mitosis. Future studies

will focus on determining how WAPL cooperates with replication stress to confer mitotic arrest; potentially through WAPL interaction with Cohesin regulators or replication factors at centromeres, or via WAPL control over chromatin architecture.

We also identified that lethal replication stress induced telomere-specific DDR signalling. Surprisingly, the mitotic telomere DDR induced by replication stress was resistant to overexpression of the telomere replication regulator TRF1. Instead, the telomere DDR resulted from activation of non-canonical mitotic telomere deprotection. Telomeres sequester the chromosome terminus within a t-loop, which when opened in a linear conformation exposes the DNA end as an ATM activating substrate[32]. We conclude human telomeres signal mitotic abnormalities induced by genomic DNA replication stress through alteration of telomere structure to activate DDR signalling. This discovery expands the role for telomeres in regulating genome stability to include the selective removal of cells with excessive genomic DNA replication stress from the cycling population.

Several lines of evidence connect mitotic telomere deprotection to Type 2 mitotic death and indicate that mitotic telomere deprotection and cohesion fatigue progress independently during mitotic arrest. Specifically, WAPL depletion imparted no significant impact on Type 2 mitotic death, indicating that Type 2 cell death was independent of cohesion fatigue. In agreement, augmenting telomere protection by TRF2 overexpression or depletion affected mitotic death in replication stressed cells without impacting cohesion phenotypes. Further, the subtle effect on mitotic death by altering TRF2 expression is consistent with telomeres contributing to the minor Type 2 pathway. Our interpretation is that cohesion fatigue and telomere deprotection occur in parallel during mitotic arrest. This is supported by our observation of numerous DDR-positive telomeres in cytocentrifuged mitotic spreads with completely separated sister chromatids. We suggest mitotic death occurs when either cohesion fatigue or telomere deprotection reaches a signalling threshold within an individual mitosis. Amplifying telomere deprotection therefore shortens the mitotic duration to cell death in replication stressed cells as the aggregate signalling from deprotected telomeres increases. Conversely, suppressing telomere deprotection does not extend mitotic duration when cohesion fatigue remained active to induce lethality.

In response to lethal replication stress, we found that suppressing apoptosis resulted in mitotic slippage and multilobular nuclei, whereas WAPL depletion elevated micronuclei. Both outcomes rescue cell death at the cost of chromosome segregation errors. Normally, supernumerary chromosomes induce a p53-dependent growth arrest[52]. However, p53 inhibition was requisite for mitotic entry in replication stressed cells. We expect

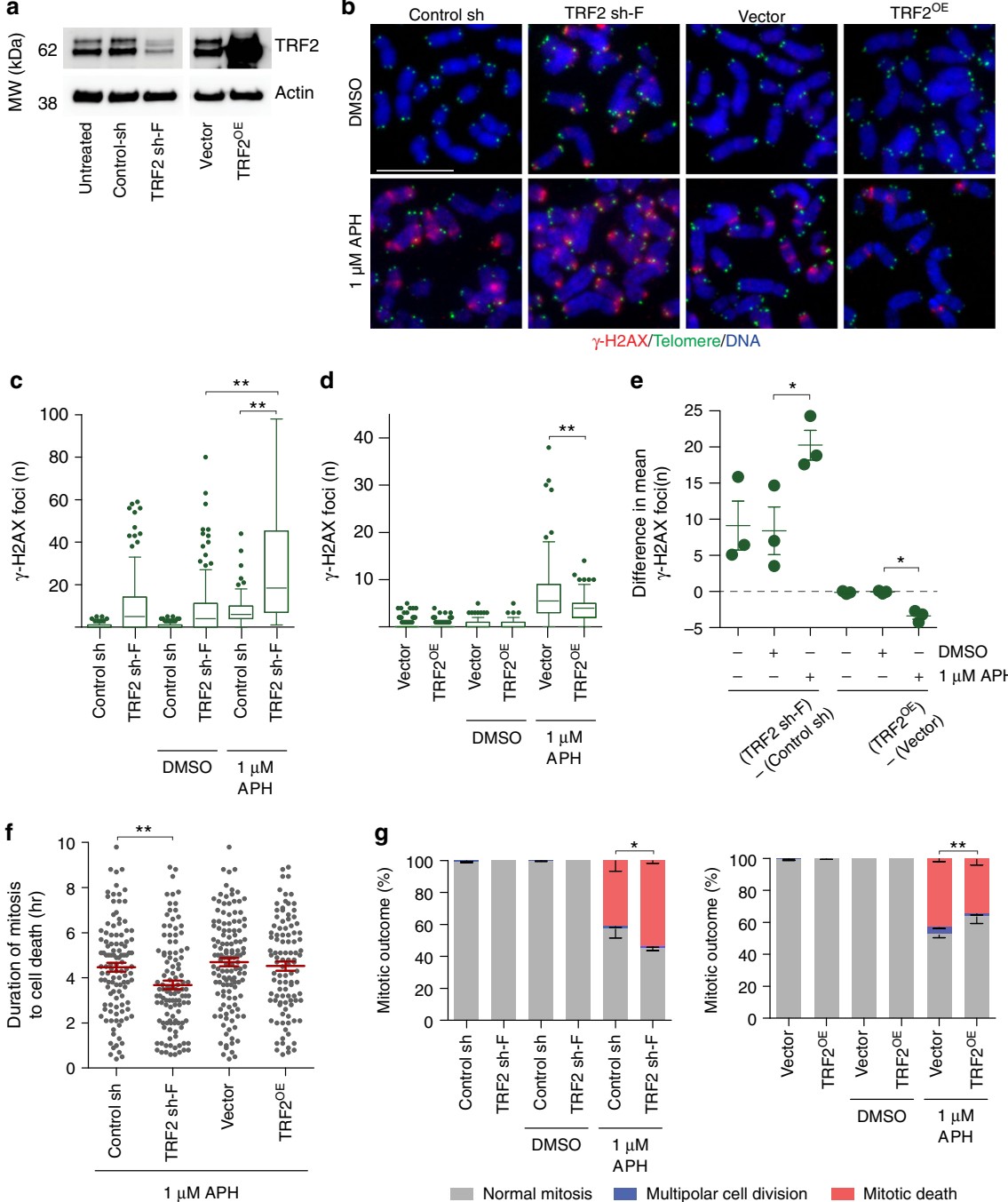

**Fig. 7** Telomere deprotection contributes to replication stress lethality. **a** Western blots of whole cell extracts from HT1080 6TG cells stably transduced with control, TRF2 shRNA (TRF sh-F) or TRF2 over expression (TRF2$^{OE}$) vectors. **b** Representative images of cyto-centrifuged chromosome spreads stained with DAPI (blue), γ-H2AX IF (red) and telomere PNA (green) from control, TRF sh-F or TRF2$^{OE}$ HT1080 6TG cells treated with DMSO or APH. Scale bar represents 10 μm. **c, d** Quantitation of mitotic telomere DDR foci from Control sh and TRF2 sh-F cells (**c**) or vector and TRF2$^{OE}$ cells (**d**) ± DMSO or APH (three biological replicates scoring $n = 50$ mitotic spreads per replicate compiled in a Tukey box plot, Mann–Whitney test). **e** Difference in the mean number of mitotic telomeric γ-H2AX foci between HT1080 6TG TRF2 sh-F or TRF2$^{OE}$ cells and their appropriate vector control. These are a different representation of the same data shown in (**c, d**) (mean ± s.e.m., $n = 3$ three biological replications, Student's $t$-test). **f** Mitotic duration to cell death in APH treated control, TRF2 sh-F and or TRF2$^{OE}$ cells (three biological replicates scoring ≥267 mitotic death events per condition are shown in a dot plot, mean ± s.e.m., Mann–Whitney test). **g** Mitotic outcome of control, TRF2 sh-F and TRF2$^{OE}$ cells treated with APH or DMSO (mean ± s.e.m., $n = 3$ biological replicates of at least 267 mitoses per condition, Fisher's Exact Test). For all panels, *$p < 0.05$, **$p < 0.01$. Source data are provided as a Source Data file

mutations that enable p53-compromised cells to avoid mitotic death will impart a substantial selective advantage in response to replication stress-inducing chemotherapeutics. While WAPL mutations are observed in cancers, they occur at a lower frequency than mutations in other cohesin subunits[53]. Notably,

STAG2 is commonly mutated in human malignancies, and some patient derived *STAG2* mutations confer reduced interaction between the STAG2 and WAPL proteins[53]. We predict mutations that rescue mitotic death by altering cohesion are more insidious than mutations inhibiting apoptosis. Multilobular nuclei

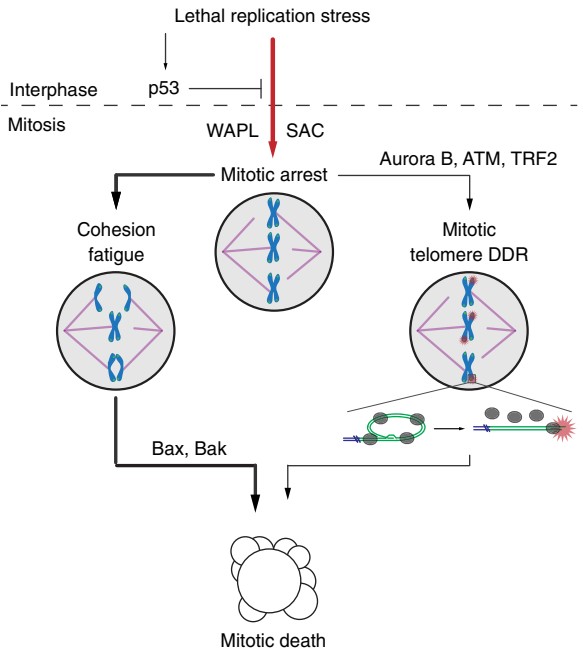

**Fig. 8** Model of replication stress-induced mitotic death. In p53-competent cells replication stress prevents mitotic entry, while lethal replication stress induces SAC-dependent mitotic arrest in cells lacking p53 function. With lethal replication stress WAPL promotes mitotic arrest which drives a dominant pathway of cell death through cohesion fatigue and BAX/BAK apoptosis. Non-canonical mitotic telomere deprotection contributes to a minority of mitotic lethality

persisting in *BAX BAK* DKO cells after APH treatment are likely incompatible with efficient proliferation. However, chromosome segregation errors that induce micronuclei, such as those occurring with rescue of mitotic lethality upon WAPL depletion, are expected to promote chromosome instability and oncogenic evolution[54]. It will be interesting to determine in the future if mutations in WAPL or its regulatory partners confer resistance to mitotic death and if this has potential clinical value in cancer stratification.

## Methods

**Cell culture and treatments**. HT1080 6TG cells were a kind gift from Eric Stanbridge (University of California, Irvine) and Saos-2 were provided by Roger Reddel (CMRI). IMR90 cells were purchased from ATCC and IMR90 E6E7 derived in the Karlseder laboratory. Phoenix cells were purchased from ATCC. Identity of all cell lines were verified by Cell Bank Australia using short tandem repeat profiling. HT1080 6TG, HeLa, Saos-2 and derivatives were cultured at 37 °C, 10% $CO_2$, and atmospheric oxygen, in DMEM (Life Technologies) supplemented with 1% non-essential amino acids (Life Technologies), 1% Glutamax (Life Technologies), and 10% bovine growth serum (Hyclone). IMR90 and derivatives were cultured at 37 °C, 10% $CO_2$, and 3% $O_2$, in DMEM, supplemented with 1% non-essential amino acids, 1% Glutamax, 10% fetal bovine serum (Life Technologies). The following compounds were used in cell treatments: dimethyl sulfoxide (DMSO, Sigma-Aldrich), colcemid (Life Technologies), Taxol (Sigma), reversine (Selleck Chemicals), KU-55933 (Calbiochem), Hesperadin (Selleck chemicals), Aphidicolin (Sigma-Aldrich) and Hydroxyurea (Sigma-Aldrich). All cell lines were tested for mycoplasma contamination (MycoAlert, LT07-118, Lonza) and were found to be negative.

**Viral transduction and cell line generation**. High titre, purified pLKO.1 derived lentiviral vectors, harbouring a non-targeting control shRNA (a gift from David Sabatini, addgene plasmid #1864), TRF2 sh-F (Open Biosystems, TRCN0000004811), or TRF2 sh-G (Open Biosystems, TRCN0000018358), were created in the Salk Institute Gene Transfer, Targeting and Therapeutics (GT3) core, or the CMRI Vector and Genome Engineering Facility[30,55]. Cell cultures were transduced with a MOI of 10 for 48 h, then selected in normal growth media supplemented with 1 μg ml⁻¹ Puromycin for 5 days. mVenus-hGeminin (1/110)/pCSII-EF and mCherry-hCdt1(30/120)/pCSII-EF (a kind gift from Atsushi

Miyawaki) were individually packaged into lentivectors using 2nd generation packaging system, and the viral supernatants were used simultaneously to co-infect target cells. 3 days post-transduction, cell cultures were sorted at the Salk Institute flow cytometry core for mVenus fluorescence, allowed to expand for 5 to 7 days, and sorted again for mCherry fluorescence. Proper progression of red and green coloration during cell cycling was confirmed with live cell imaging before use in experimentation.

Retroviral vectors were created by transfecting Phoenix-AMPHO cells with pWZL-hygro (Vetctor, a gift from Scott Lowe, addgene plasmid #18750), pWZL Hygro-TRF2 (TRF2^OE, a gift from Titia de Lange, addgene plasmid #16066), pWZL-hTERT (hTERT from pBabe-hTERT, a gift from Titia de Lange, was recloned into pWZL-hygro), pWZL-H2B-mCherry[55], pLPC-mCherry-TRF1 [mCherry was cloned into pLPC-NFLAG TRF1 (a gift from Titia de Lange, addgene plasmid #16058) by PCR amplification using BamHI restriction sites. Forward primer: 5′-CGG GGATCC ATG GTG AGC AAG GGC GAG GAGG-3′/ Reverse primer: 5′-CGG GGATCC GGT GGC GAT GCT GCG CTT GTA CAG CTC GTC CAT GCC-3′ A linker was added in C-terminus of mCherry], pLXSN3 or pLXSN3-p53DD[38]. Viral supernatants were used to infect HT1080 6TG or IMR90 cells. Stable vector and TRF2^OE cell lines were selected by treatment with 100 μg ml⁻¹ Hygromycin (Invitrogen). HT1080 6TG H2B-mCherry cells were sorted for mCherry fluorescence at the Salk Institute flow cytometry core facility. IMR90 TERT were selected with 100 μg ml⁻¹ Hygromycin, followed by second infection with retrovirus carrying pLXSN3 or pLXSN3-p53DD and selection with 600 μg ml⁻¹ G418 (Nacalai Tesque).

**CRISPR/Cas9 gene targeting**. *BAX BAK* DKO HeLa, HT1080 6TG, and HT1080 6TG H2B-mCherry were generated by transiently transfecting with pSpCas9(BB)-2A-GFP (PX458) plasmids[56] (a gift from Feng Zhang; Addgene plasmid #48138) harbouring guide RNAs targeting human BAX (5′-CTGCAGGATGATTGCCGC CG) or human BAK (5′-GCATGAAGTCGACCACGAAG) according to the manufacturer's protocol (XtremeGene; Sigma-Aldrich). Individual clones were initially screened for *BAX* and *BAK* deletion by immunoblotting, and then confirmed by sequencing. Because of their limited proliferation we did not select clonal IMR90 knockout cells. Instead, we co-transduced a population of IMR90 fibroblasts with a mCherry-Cas9 using FUCas9Cherry (a kind gift from Marco Herold) and a pool of pKLV-U6gRNA(BbsI)-PGKpuro2ABFP (a gift from Kosuke Yusa; Addgene plasmid #67974) lentivectors[57] carrying six p53 gRNA sequences (5′-CCCCGGACGATATTGAACAA, 5′-GAGCGCTGCTCAGATAGCGA, 5′-TG GCCATCTACAAGCAGTCA, 5′-ACTCGGATAAGATGCTGAGG, 5′-GATCCA CTCACAGTTTCCAT, and 5′-GGTGCCCTATGAGCCGCCTG). Cells were FACS sorted for BFP and mCherry double positive cells and cultured as above.

**siRNA transfection**. Control non-targeting (control siRNA, D-001810-10) and WAPL (L-026287-01) ON-TARGETplus siRNA pools (Dharmacon) were transfected using Lipofectamine RNAi max (Thermofisher Scientific), according to the manufactures' protocols.

**Live cell imaging and analysis**. DIC microscopy was used to visualize mitotic duration and outcome. These experiments were performed on a Zeiss Cell Observer inverted wide field microscope, with 20 × 0.8 NA air objective, at 37 °C, 10% $CO_2$ and atmospheric oxygen. Images were captured every 6 min for a duration up to 65 h using an Axiocam 506 monochromatic camera (Zeiss) and Zen software (Zeiss). FUCCI live cell imaging was conducted on the same instrument, using the same imaging duration and protocol, with the addition of a Zeiss HXP 120C mercury short-arc lamp and compatible filter cubes to obtain fluorescent images. Quantitative live cell imaging of mitotic chromosome dynamics in HT1080 6TG H2B-mCherry cultures and derivatives was done using either combined DIC and fluorescent imaging on the Zeiss Cell Observer described above with a 40 × 0.95 NA plan-Apochromat air objective; or on a Zeiss Cell Observer SD spinning disk confocal microscope imaged using a 561 nm, 50 mW solid state excitation laser and appropriate filter sets with a 40×, 1.3 NA objective and an Evolve Delta camera (Photometrics). For these experiments, cells were cultured at 37 °C, 10% $CO_2$ and atmospheric oxygen and images captured every 3 min for 60 h. For all movies, mitotic duration was scored by eye and calculated from nuclear envelope breakdown until cytokinesis or mitotic death. FUCCI Movies were scored by eye, for G1 (red) and S/G2 (green) by colour, and for mitotic entry, duration and outcome by cell morphology. Chromosome dynamics in HT1080 6TG-mCherry H2B were scored by eye. All live cell imaging analysis was done using Zen software.

**Visualization of cohesion status and mitotic DDR activation**. For cohesion analysis, standard methanol and acetic acid fixed cytogenetic chromosome spreads were prepared and stained with fluorescent in situ hybridization (FISH) with a telomere (Alexa488-OO-ccctaaccctaaccctaa), and centromere (Texas Red-OO-aaactagacagaagcatt), peptide nucleic acid (PNA) probes (Panagene), by denaturing at 80 °C, followed by overnight hybridization in a dark humidity chamber[58]. The mitotic telomere DDR was visualized using the metaphase-TIF assay[58]. In this method, cells were cytocentrifuged onto glass slides, fixed and stained with telomere PNA FISH and immunofluorescence (IF) against γ-H2AX. For both assays, images were captured on a Zeiss AxioImager Z.2 with a 63×, 1.4 NA oil objective,

appropriate filter cubes and a CoolCube1 camera (Metasystems). Automated metaphase finding and image acquisition for these experiments were done using the MetaSystems imaging platform[36].

**Cytogenetic and interphase image analysis.** Cytogenetic images were analyzed using Isis software (MetaSystems). For cohesion analysis, images were scored by eye, and mitotic spreads classified as follows: normal, if <3 chromosomes displayed visible separated sister chromatids; mild, if ≥3 chromosomes displayed visible sister chromatids; moderate if ≥3 chromosomes displayed centromere cohesion loss but retained distal chromosome arm cohesion; severe if ≥3 chromosomes displayed complete sister chromatid separation. In practice, we typically observed the majority of chromosomes in a mitotic spread displayed the same cohesion status. Mitotic DDR foci were quantified by eye using Isis software, and classified as telomeric if the γ-H2AX IF focus colocalized with a telomere PNA signal. To determine relative telomere lengths at DDR(+) and DDR(−) chromosome ends, telomere signals were captured at fixed exposure time as described above, and images were analyzed using Imaris software version 8.2 (BitPlane). Images were segmented using the mitotic chromosomes as a region of interest, and the telomeres were detected as spots using the spot detection wizard. Telomere colocalization with γ-H2AX foci were identified using the co-localisation tool. The sum fluorescence intensity was determined for each telomere in the γ-H2AX colocalised (DDR+) and the γ-H2AX excluded (DDR−) cohorts. To assay interphase nuclei for micronuclei and multilobular cells, cultures were grown, treated and fixed on Alcain blue treated glass coverslips, stained with DAPI, washed, mounted and imaged using a on a Zeiss AxioImager Z.2 with a 63× 1.4 NA oil objective using Zen software. Images were scored by eye.

**Chromatin fibre analysis.** Replication rates were measured using chromatin fibre analysis[59]. Briefly, unsynchronized cells are sequentially pulse-labelled with 100 μM of thymidine analogues (IdU then CldU) for 30 min each. After harvesting, the genomic DNA was stretched as fibres onto glass slides at a constant rate of 2 kb μm⁻¹ using a molecular combing system (Genomic Vision). Nascent DNA replication was visualized by IF against IdU and CldU and captured using Zen software and a Zeiss AxioImager Z.2, with a 63×, 1.4 NA oil objective, appropriate filter cubes and an Axiocam 506 monochromatic camera (Zeiss). In our analysis, only replication forks with a visible origin of replication, and both IdU and CLdU staining were scored. Replication rates were calculated based solely on the IdU tracks, which result from the first pulse of nucleoside analogue. The length of IdU fibres were converted to kilobasepairs according to a constant and sequence-independent stretching factor (1 μm = 2 Kb) using Zen software.

**Western blotting.** Preparation of whole cell extracts and western blots were performed[32] and luminescence was visualized on an LAS 4000 Imager (Fujifilm).

**Antibodies.** Primary antibodies used in this study: γ-H2AX (1:1000, 05-636, Millipore), TRF1 (1:500, sc-56807, Santa Cruz), TRF2 (1:1000, NB110-57130, Novus Biologicals), WAPL (1:1000, SC-365189, Santa Cruz), β-actin (1:10,000, A5441, Sigma), BAX (1:1000, rat monoclonal 49F9; WEHI monoclonal antibody facility), BAK (1:2000, rabbit polyclonal B5897, Sigma), CldU (1:25, 6326, Abcam), IdU (1:5, 347580, BD Biosciences), p53 (p53 (DO-1): 1:200, sc-126, Santacruz), p53-ser10 (1:1000, 9284S, Cell Signalling), Vinculin (1:10,000, V9131, Sigma). Secondary antibodies used in this study: Alexa Fluor 568-Goat anti-mouse IgG (1:1000, A-11031, Thermo Fisher Scientific), Alexa Fluor 488-Goat anti-mouse (1:1000, A11029, Thermo Fisher Scientific) and Alexa Flour 594-Goat anti-rat IgG (1:1000, A-11007). Goat anti-Rabbit HRP (1:1000, P0448, Dako), Goat anti-Mouse HRP (1:1000, P0447, Dako).

**Statistics and Figure preparation.** Statistical analysis was performed using GraphPad Prism. Box plots are displayed using the Tukey method where the box extends from the 25th to the 75th percentile data points and the line represents the median. The upper whisker represents data points ranging up to the 75th percentile + (1.5 × the inner quartile range), or the largest value data point if no data points are outside this range. The lower whisker represents data points ranging down to the 25th percentile− (1.5 × the inner quartile range), or the smallest data point if no data points are outside this range. Data points outside these ranges are shown as individual points. Error bars, statistical methods and n, are described in figure legends. In Fig. 5d statistical significance was determined by a t-test for all conditions expect mitotic slippage, where a Fisher's exact test was applied. This is because a t-test cannot be used when all outcomes are zero for two of the conditions. Figures were prepared using Adobe Photoshop and Illustrator.

**Reporting summary.** Further information on research design is available in the Nature Research Reporting Summary linked to this article.

## Data availability
The data that support the findings of this study are archived at the Children's Medical Research Institute, Kyoto University, or the Walter and Eliza Hall Institute of Medical

Research, and are available from the corresponding author upon reasonable request. The source data underlying all the presented western blots, histograms, and dot plots are available in the Source Data file.

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

## Acknowledgements

Scott Page and the CMRI ACRF Telomere Analysis Centre, supported by the Australian Cancer Research Foundation, are thanked for microscopy infrastructure. Leszek Lisowski is thanked for generating viral vectors at the Salk Institute's GT3 Core and CMRI's Vector and Genome Engineering Facility. The Salk Institute's Flow Cytometry Core is thanked for assistance with cell sorting. V.P.M. is supported by an Australian Post-Graduate Award from the University of Sydney. N.L. is supported by fellowships from the Hebrew University Smorgon Foundation and the Cancer Institute NSW. L.C. is supported by the ATIP starting grant from CNRS in the framework of Plan Cancer 2014-2019, the ANR Tremplin ERC (55911) and the European Research Council grant telo-HOOK/ERC (714653). J.K. is supported by The NIH (R01GM087476, R01CA174942), the Donald and Darlene Shiley Chair, the Highland Street Foundation, the Fritz B. Burns Foundation, the Emerald Foundation and the Glenn Center for Aging Research. Work in the laboratory of D.C.S.H is supported by fellowships and grants from the Australian NHMRC (1043149, 1016701, 1113133), Leukemia Lymphoma Society SCOR (11283-17), Independent Research Institutes Infrastructure Support Scheme grant (9000220) and a Victorian State Government Operational Infrastructure Support (OIS) grant. M.T.H. is supported by grants from the Senri Life Science Foundation, the Uehara Memorial Foundation, the Daiichi Sankyo Foundation of Life Science, the Nakajima Foundation, The Mochida Memorial Foundation Research Grant, Grant-in-Aid for Young Scientists (A) (16H06176) and Grant-in-Aid for Scientific Research on Innovative Areas (16H01406). A.J.C. is supported by grants from the Australian NHMRC (1106241, 1104461, 1162886), the Cancer Council NSW (RG 15-12), the Cancer Institute NSW (11/FRL/5-02), and philanthropy from Stanford Brown, Inc (Sydney, Australia) and the Goodridge Foundation.

## Author contributions

V.P.M., J.K., D.C.S.H., M.T.H. and A.J.C. conceived the study and designed experiments. V.P.M, R.R.J.L., K.S.M., A.O'C., C.D.R., N.L., L.C. and M.T.H. performed experiments. V.P.M. and A.J.C. wrote the manuscript with editorial assistance from J.K., D.C.S.H., and M.T.H.

## Additional information

**Competing interests:** The authors declare no competing interests.

