## [Peer Review File · Nature Communications]

Reviewers' comments:

Reviewer #1 (Remarks to the Author):

This manuscript by Masamsetti et al. addresses how replication stress induces mitotic lethality in human cells. Using live imaging of single cells to track cell cycle progression and mitotic outcomes following lethal dosages of replication stress treatments (aphidicolin and hydroxyurea), the authors reported that continuous replication stress causes prolonged mitoses and mitotic death specifically in p53 compromised cells. Importantly, as detected with FUCCI coupled live imaging, the majority of the mitotic death occurs immediately in the first mitosis, indicating replication stress is effective in driving mitotic catastrophe. Chromosome spreads from replication stressed p53 compromised cells reveal chromosome phenotypes that are associated with aberrant chromosome cohesion. Interestingly, depletion of cohesion antagonist, WAPL, not only rescues the aberrant chromosome cohesion phenotypes, but also significantly reduces mitotic duration and cell death, providing evidence that prolonged mitosis/mitotic arrest from spindle assembly checkpoint (SAC) activation drives cohesion fatigue and mitotic cell death. Though exactly how WAPL signals to BAX/BAK-dependent apoptosis remain to be investigated, depletion of both BAX and BAK similarly leads to reduced mitotic death (but increased mitotic duration in Supp Fig 3c-d). In addition to the primary cell death events that are dependent on WAPL and BAX/BAK, there is also a secondary mitotic death pathway regulated by TRF2, Aurora B and ATM.

This work synthesizes known concepts, and while perhaps not groundbreaking, does provide important information that will be of interest to the field. Importantly, the study is nicely executed, well-controlled and leads to some interesting conclusions. I have only two major comments and a number of minor comments:

Major comments:

1) The authors establish that BAX/BAK is responsible for the majority of replication stress induced mitotic death. While double knockout of BAX/BAK suppresses cell death, the mitotic duration increases. This can be interpreted (according to the conclusions from this paper) as cells experience SAC dependent mitotic arrest but eventually complete mitosis and escape death. If this is true, one would expect to see chromosome cohesion defects using cytogenetic chromosome spreading when these double knockout cells are treated with APH or HU. Do the authors observe such phenotypes?

2) Type 1 mitotic death is classified based on observable dispersion of chromosomal material prior to lethality and is concluded to be dependent on WAPL and BAX/BAK. Is the dispersion of chromosomal material the consequence of massive sister chromatid detachment?

Minor comments:

1) The status of p53 is key to determine replication stress-driven mitotic outcomes as described in this paper. Hence, it is worthwhile to have a sentence or two to describe current knowledge of p53 functions in cell cycle checkpoint and/or apoptosis. This may help to rationalize why the authors were comparing p53-competent versus -deficient cells.

2) There are a few typographical errors such as:

- a. Page 5, line 8: "in a significant increases in" should be in a significant increase in
- b. Page 10, line 7: the period at the end of the line should be a comma

Reviewer #2 (Remarks to the Author):

The manuscript by Masamsetti et al. attempt to explain the mechanism by which cells die during mitotic catastrophe after experiencing replication stress. Using live-cell imaging the authors show that cells die via mitotic catastrophe after a prolonged exposure to “lethal” dose of aphidicolin and hydroxyurea, two replication stress agents. The authors characterize this in primary IMR90 and E6E7 transformed IMR90 fibroblasts, and p53-impaired HCT1080 6TG, HeLa, and Saos-2 cells. Using the FUCCI system, the authors go on to show that the observed mitotic death occurs in the same cell cycle as when a lethal dose of Aph/HU is given, showing daughter cells are not inheriting damage from a previous cycle and then succumbing to mitotic death. The bulk of the manuscript then goes on to characterize the mitotic death, and split it into 3 distinct modes of death, Type 1, Type 2, and death during interphase. The authors show that type 1 is characterized by chromosome dispersal preceding death and that this process is driven by cohesion fatigue. Interestingly, when the authors depleted either WAPL (antagonist of cohesion) or BAX/BAK in p53-deficient HT1080 cells they were able to prevent Type 1 mitotic death, reasoning that cohesion fatigue drives the death in a WAPL and BAX/BAK-dependent manner. Type 2 mitotic death is characterized by chromosome collapse without the chromosome dispersal. This was attributed to a mitotic telomere deprotection dependent on the shelterin component, TRF2, dissociating from telomeres which then induced DDR.

The authors present compelling evidence for p53-dependent mitotic cell death after high dose replication stress, using a combination of live-cell imaging and cytogenetic analysis. However, the characterization of type 1 death remains fairly descriptive and it is unclear how the cohesion fatigue induces apoptosis.

Major points:

1. Considering that E6 and E7 target both p53 and pRb and the other cell lines are highly mutated and/or highly aneuploidy, it would be cleaner to use CRISPR p53 KO in primary cells to show clear p53 dependency of the mitotic death.
2. Mitotic death may be occurring due to under-replication after the high doses of Aphidicolin and HU. Does treatment with RO-3306 rescue the phenotype? Is there excess ssDNA at the G2/M transition and this is the cause of death for any of the 3 types of death? Authors should also include cell cycle analysis with BrdU and PI to show the extent of S phase progression before entry into mitosis. Additionally, do lower doses of Aph/HU cause any shift in the type of mitotic death that occurs?
3. Although mitotic death is decreased upon WAPL depletion it is unclear that WAPL is truly driving the mitotic death. These experiments implicate the cohesion pathway but do not address if WAPL is actively involved. For the WAPL depletion experiments, do authors observe the “vermicelli” phenotype described in previous studies of WAPL? Does it affect their quantification of phenotypes?
4. It is not explored if and how WAPL and BAX/BAK genetically interact. It is possible that they just phenocopy each other. Have the authors depleted WAPL in the DKO cells?
5. It has been recently reported that pericentromeric replication requires TRF2. It appears from chromosome spreads that gammaH2AX may be enriched at centromeres not just at telomeres. If that is the case, the type 2 mitotic death could be a mixture of both under-replication and telomere uncapping.
6. Authors should refer to the following studies as they are pertinent to their manuscript.

-Haarhuis et al. 2013. Current Biology. WAPL-Mediated Removal of Cohesin Protects against

Segregation Errors and Aneuploidy

-Mendez-Bermudez et al. 2018. Genome-wide Control of Heterochromatin Replication by the Telomere Capping Protein TRF2

Reviewer #3 (Remarks to the Author):

In this work, the authors investigate the cellular response to DNA replication stress. They employ Aphidicolin (APH) or Hydroxyurea (HU) to induce replication stress and visualize cellular outcome using live-cell imaging. They find that in p53-deficient cells, replication stress induces a spindle assembly checkpoint (SAC)-dependent mitotic arrest and cell death. They found that this mitotic death is triggered by the activation of two pathways: BAX/BAK-mediated apoptosis and a cohesion defect that is mediated by WAPL, and a secondary pathway triggered by telomere deprotection. They also provide evidence that suppressing mitotic death promotes genome instability in cells that experienced replication stress. The data in this study provide mechanistic insight into the pathways that upon replication stress induce mitotic catastrophe. This is useful information that may guide future experiments with implication for cancer treatment and defining the pathways that ensure genome stability.

1. The data in support of a relationship between suppression of mitotic cell death and induction of genome instability is not entirely convincing/clear. The authors provide few data in support of genome instability being an outcome of replication stress when mitotic cell death is inhibited. Moreover, the authors show that blocking mitotic death in two independent manners causes very different effects: WAPL inhibition, when combined with APH treatment, conferred an increase in multipolar cell division and micronuclei (Fig. 4j,k). In contrast, APH treatment in BAX BAK DKO cells did not result in the same effect. This is puzzling given that the authors concluded that WAPL and BAX/BAK act on the same pathway (Supplemental Fig. 5). Additional experiments are required to clarify what are the consequences of replication stress when mitotic death is inhibited.

We thank the reviewers for their comments and constructive feedback which have improved the manuscript. Please see our responses below in blue text.

As a result of revision, the following changes were made to the manuscript.

- Fig. 3e includes new data.
- The previous Fig. 4 was split. Panels Fig. 4a-g remain in Fig 4. The Panels that were Fig. 4h-k are now Fig. 5 a-d. New data is included as Fig 5e-j.
- Previous Figs. 5-7 are now Figs. 6-8.
- The previous Supplementary Fig. 1 was split. Panels Supplementary Fig. 1a-c remain, with new data included as Supplementary Fig. 1d-g. The previous panels that were Supplementary Fig 1d-g are now Supplementary Fig 2a-d.
- Previous Supplementary Figs. 2-7 are now Supplementary Figs. 3-8.
- A new panel was added as Supplementary Fig. 8g.
- A new movie was added as Supplementary Movie 4.

Minor manuscript revisions to the text made to accommodate movement of figure panels are not highlighted with different coloured text. Revisions that directly address reviewers' comments as indicated below are highlighted in the manuscript with red text.

As a result of new experimentation and analysis, Ronnie Low was moved from 3rd to 2nd author, and Aisling O'Connor was added to the author list. All authors agree on the changes. The methods were also edited to include additional information related to p53 CRISPR and imaging experiments.

Reviewers' comments:

Reviewer #1

This manuscript by Masamsetti et al. addresses how replication stress induces mitotic lethality in human cells. Using live imaging of single cells to track cell cycle progression and mitotic outcomes following lethal dosages of replication stress treatments (aphidicolin and hydroxyurea), the authors reported that continuous replication stress causes prolonged mitoses and mitotic death specifically in p53 compromised cells. Importantly, as detected with FUCCI coupled live imaging, the majority of the mitotic death occurs immediately in the first mitosis, indicating replication stress is effective in driving mitotic catastrophe. Chromosome spreads from replication stressed p53 compromised cells reveal chromosome phenotypes that are associated with aberrant chromosome cohesion. Interestingly, depletion of cohesion antagonist, WAPL, not only rescues the aberrant chromosome cohesion phenotypes, but also significantly reduces mitotic duration and cell death, providing evidence that prolonged mitosis/mitotic arrest from spindle assembly checkpoint (SAC) activation drives cohesion fatigue and mitotic cell death. Though exactly how WAPL signals to BAX/BAK-dependent apoptosis remain to be investigated, depletion of both BAX and BAK similarly leads to reduced mitotic death (but increased mitotic duration in Supp Fig 3c-d). In addition to the primary cell death events that are dependent on WAPL and BAX/BAK, there is also a secondary mitotic death pathway regulated by TRF2, Aurora B and ATM.

This work synthesizes known concepts, and while perhaps not groundbreaking, does provide important information that will be of interest to the field. Importantly, the study is nicely executed, well-controlled and leads to some interesting conclusions. I have only two major comments and a number of minor comments:

Major comments:

1) The authors establish that BAX/BAK is responsible for the majority of replication stress induced mitotic death. While double knockout of BAX/BAK suppresses cell death, the mitotic duration increases. This can be interpreted (according to the conclusions from this paper) as cells experience SAC dependent mitotic arrest but eventually complete mitosis and escape death. If this is true, one would expect to see chromosome cohesion defects using cytogenetic chromosome spreading when these double knockout cells are treated with APH or HU. Do the authors observe such phenotypes?

We completed the requested experiment and observed that the aberrant cohesion phenotype is present in APH treated *BAX BAK* double knock out (DKO) cells. Notably, in the *BAX BAK* DKO cells there are significantly more occurrences of the severe cohesion phenotype (completely separated sister chromatids) as compared to parental controls. We interpret these data to indicate that as a result of inhibited apoptotic pathways, *BAX BAK* DKO cells cannot eliminate mitoses following sister chromatid separation. Therefore, we observe more occurrences of severe cohesion fatigue in the population of mitotically arrest cells. These findings support our conclusion that cohesion fatigue is a precipitating event in *BAX BAK*-dependent mitotic death. The new data are presented in Figure 5g and we discuss this result and our interpretation briefly in the corresponding text on page 12 and in the discussion on page 17.

2) Type 1 mitotic death is classified based on observable dispersion of chromosomal material prior to lethality and is concluded to be dependent on WAPL and BAX/BAK. Is the dispersion of chromosomal material the consequence of massive sister chromatid detachment?

We observe separated sister chromatids in cytogenetic and cyto-centrifuged chromosome preparations from APH or HU treated cells (Fig 4 and Supplementary Fig. 7c). The phenotype is easily observed in these experiments because chromosomes are spread on a substrate prior to observation. These data are consistent with the corresponding chromosome dispersion phenotype in live-cell imaging.

We exhaustively attempted to quantitatively demonstrate that the chromatin dispersion phenotype is indeed chromatid separation through: 1) live-cell imaging of H2B-mCherry, or H2B-mCherry and GFP-TRF1 expressing cells +/- APH treatment; and 2) imaging mitotic cells in adherent asynchronous cell populations +/- APH following fixation on their growth substrate and staining with telomere and pan-centromeric probes. Under both conditions, structures on sister chromatids (e.g. sister chromatid telomeres or centromeres) were often close enough in three-dimensional space to appear as single foci in light microscopy. Thus, we could not conclusively determine when single telomere or centromere foci represented single or cohered chromatids. Telomere and pan-centromere probes also displayed heterogenous fluorescence intensity, preventing us from measuring fluorescence intensity as a proxy to determine single vs. cohered chromatids. Additionally, cells displaying the chromatin dispersion phenotype also typically displayed highly mobile mitotic chromosomes that did not align for intra-kinetochore distance measurements. We also attempted to stain individual chromosomes with specific centromere FISH probes and measure distances between foci as a proxy for chromatid separation. However, we were unable to effectively stain the condensed mitotic chromatin of replication stressed cells, resulting in uninformative experiments.

We continue to interpret our data from cytogenetic and cyto-centrifuged chromosome preparations to indicate that the chromatin dispersion phenotype represents separated sister chromatids. We only observe the cohesion phenotype and the chromatin dispersion phenotype under lethal replication stress conditions. Further, both the cohesion phenotype and the chromosome dispersion phenotype

are inhibited by WAPL depletion. We never observed evidence of broken or shattered chromatin as an alternative mechanism of chromatin dispersion in any chromosome preparations. Additionally, our new observation that inhibiting apoptosis in *BAX BAK* DKO cells worsens the cohesion phenotype is consistent with separated sister chromatids signalling the mitotic death event. This supports our assertion that the cohesion phenotype occurs in real time in mitotic cells as a result of lethal replication stress, and this is what we observe with live-cell imaging. We hope that reviewer appreciates our attempt to address this question and agrees with our conclusion.

Minor comments:

1) The status of p53 is key to determine replication stress-driven mitotic outcomes as described in this paper. Hence, it is worthwhile to have a sentence or two to describe current knowledge of p53 functions in cell cycle checkpoint and/or apoptosis. This may help to rationalize why the authors were comparing p53-competent versus -deficient cells.

We added a cause on page 5, middle paragraph, lines 2-4 to clarify our reasoning.

2) There are a few typographical errors such as:

a. Page 5, line 8: “in a significant increases in” should be in a significant increase in.

b. Page 10, line 7: the period at the end of the line should be a comma.

Thank you for bringing these errors to our attention. They have been corrected in the text.

Reviewer #2

The manuscript by Masamsetti et al. attempt to explain the mechanism by which cells die during mitotic catastrophe after experiencing replication stress. Using live-cell imaging the authors show that cells die via mitotic catastrophe after a prolonged exposure to “lethal” dose of aphidicolin and hydroxyurea, two replication stress agents. The authors characterize this in primary IMR90 and E6E7 transformed IMR90 fibroblasts, and p53-impaired HCT1080 6TG, HeLa, and Saos-2 cells. Using the FUCCI system, the authors go on to show that the observed mitotic death occurs in the same cell cycle as when a lethal dose of Aph/HU is given, showing daughter cells are not inheriting damage from a previous cycle and then succumbing to mitotic death. The bulk of the manuscript then goes on to characterize the mitotic death, and split it into 3 distinct modes of death, Type 1, Type 2, and death during interphase. The authors show that type 1 is characterized by chromosome dispersal preceding death and that this process is driven by cohesion fatigue. Interestingly, when the authors depleted either WAPL (antagonist of cohesion) or BAX/BAK in p53-deficient HT1080 cells they were able to prevent Type 1 mitotic death, reasoning that cohesion fatigue drives the death in a WAPL and BAX/BAK-dependent manner. Type 2 mitotic death is characterized by chromosome collapse without the chromosome dispersal. This was attributed to a mitotic telomere deprotection dependent on the shelterin component, TRF2, dissociating from telomeres which then induced DDR.

The authors present compelling evidence for p53-dependent mitotic cell death after high dose replication stress, using a combination of live-cell imaging and cytogenetic analysis. However, the characterization of type 1 death remains fairly descriptive and it is unclear how the cohesion fatigue induces apoptosis.

Major points:

1. Considering that E6 and E7 target both p53 and pRb and the other cell lines are highly mutated and/or highly aneuploidy, it would be cleaner to use CRISPR p53 KO in primary cells to show clear p53 dependency of the mitotic death.

Our data indicate replication stress-induced mitotic death requires loss of p53 function. Cells with a functional p53 arrest cell growth and fail to enter mitosis, therefore avoiding death. To confirm, we performed the suggested experiment and targeted p53 in IMR90 fibroblasts using CRISPR/Cas9. Because of the limited proliferative lifespan of primary IMR90 cells, we opted to target p53 on the population level and sort for transduced cells, as opposed to selecting individual clones. Western blots revealed p53 was largely eliminated from the cell population. In agreement with our previous conclusions, live cell imaging identified that the IMR90 p53 CRISPR population displayed common mitotic arrest and mitotic death in the presence of a 1 μ M APH. This confirms that loss of p53 is required for mitotic death. These new data are presented as Supplementary Fig 1d-g and addressed in the text on page 5, bottom of the second paragraph.

2. Mitotic death may be occurring due to under-replication after the high doses of Aphidicolin and HU. Does treatment with RO-3306 rescue the phenotype? Is there excess ssDNA at the G2/M transition and this is the cause of death for any of the 3 types of death? Authors should also include cell cycle analysis with BrdU and PI to show the extent of S phase progression before entry into mitosis. Additionally, do lower doses of Aph/HU cause any shift in the type of mitotic death that occurs?

Unfortunately, treatment with the CDK1 inhibitor RO-3306 induces growth arrest at the G2/M boundary and prevents mitotic entry (Vassilev et al, 2006 PNAS). Because RO-3306 prevents mitosis entry, we cannot test the impact of this compound on mitotic death as treated cells will not progress into cell division in the presence of the inhibitor. We did test how lower doses of APH impacts mitotic death in HT1080 6TG cells and observed a similar distribution in Type 1 vs Type 2 death in 1.0 and 0.75 μ M APH treated cells (Data added to Fig. 3e). The notable difference was that more interphase cell death occurred with the higher dose of 1.0 μ M APH. This indicates that lethal replication stress primarily induces mitotic death. However, increasing amounts of replication stress above a lethal dose, correlating with more under-replicated DNA, leads to a minor but increased occurrence of interphase death.

We note that cytogenetic and cyto-centrifuged chromosome preparations both revealed that mitotic chromosomes in APH or HU treated cells display fully formed sister chromatids. This indicates that even in the presence of lethal dosages of replication stress inducing drugs, DNA replication is largely completed prior to mitotic entry. To address the extent of S-phase progression before entry into mitosis we performed H3-S10P

Response Figure 1: Flow cytometry measuring DNA content and mitotic cells in HT1080 6TG cultures treated for 48 hours with vehicle or 1 μ M APH. Mitotic cells are identified with an anti-H3 phospho-S10 antibody. The data show overlapping DNA content in mitotic cells from APH and vehicle treated cultures. We note the presence of tetraploidy in this cell population, which is common for HT1080 and derivative cell lines.

flow cytometry (instead of BrdU and PI staining) to visualize bulk DNA content in mitotic cells treated with DMSO or 1 μ M APH (Response Figure 1). This further suggests DNA replication is largely complete upon mitotic entry in our experimental conditions. Additionally, we show that APH washout during mitosis, which enables mitotic DNA synthesis to fill-in under-replicated regions (Minocherhomji et al, Nature 2015), fails to rescue mitotic death. Finally, mitotic death was dependent on mitotic arrest, and both cohesion fatigue and mitotic telomere deprotection are dependent upon mitotic arrest for their occurrence. We do not exclude that under-replicated DNA occurs with replication stress. However, we conclude the preponderance of data suggest that DNA under-replication is not principally involved in promoting mitotic death.

3. Although mitotic death is decreased upon WAPL depletion it is unclear that WAPL is truly driving the mitotic death. These experiments implicate the cohesion pathway but do not address if WAPL is actively involved. For the WAPL depletion experiments, do authors observe the “vermicelli” phenotype described in previous studies of WAPL? Does it affect their quantification of phenotypes?

We did observe that almost all interphase nuclei displayed the vermicelli phenotype in WAPL depleted cultures (Response Figure 2) and contend this observation does not impact our quantification of mitotic outcomes. In the discussion we expound that WAPL function in replication stressed interphase cells promotes the minor cohesion fatigue phenotype, which leads to subsequent mitotic arrest and mitotic death in the same cell

cycle. This is consistent with interphase function dictating later mitotic outcomes. We also now show that depleting WAPL in *BAX BAK* DKO cells enables replication stressed cultures to accumulate more mitotic cells with completely separated sister chromatids. This argues that complete cohesion fatigue, regulated by WAPL, promotes mitotic death through apoptosis. WAPL may not drive mitotic death per se, but WAPL is regulating altered cohesion, which promotes mitotic arrest and mitotic death. We maintain the data are consistent with our interpretation and discussion of mitotic death regulated by WAPL.

4. It is not explored if and how WAPL and BAX/BAK genetically interact. It is possible that they just phenocopy each other. Have the authors depleted WAPL in the DKO cells?

We agree with Reviewer #2 and Reviewer #3 that a better description on the functional relationship between WAPL and BAX/BAK is required. To address this, we added new data in Fig 5e-g, including the requested experiment of WAPL depletion in *BAX BAK* DKO cells. We also added the accompanying paragraph in the result section on page 12 and 13, and text in the discussion on page 17 top and bottom paragraph, to better explain how WAPL and BAX/BAK function in the cell death pathway.

Our data are consistent with a pathway where 1) lethal replication stress induces a WAPL-dependent minor cohesion phenotype leading to subsequent SAC-dependent mitotic arrest; 2) mitotic arrest then promotes cohesion fatigue; and 3) cohesion fatigue then results in BAX/BAK-dependent apoptosis. Depleting WAPL in *BAX BAK* DKO cells rescued mitotic duration, consistent with our interpretation of

WAPL functioning upstream of mitotic arrest (Fig. 5e, f). Assaying cohesion status in APH treated *BAX BAK* DKO cells revealed worsening cohesion fatigue, consistent with cohesion fatigue signalling apoptosis (Fig. 5g). Apoptosis suppressed cells thus accumulate more mitotic cells with completely separated sister chromatids. Our interpretation is that WAPL and *BAX BAK* have independent functions occurring in temporal order. WAPL functions upstream to regulate the initiating mitotic arrest, and *BAX/BAK* downstream as the executor promoting the terminal consequence of mitotic death.

5. It has been recently reported that pericentromeric replication requires TRF2. It appears from chromosome spreads that γ H2AX may be enriched at centromeres not just at telomeres. If that is the case, the type 2 mitotic death could be a mixture of both under-replication and telomere uncapping.

We assayed γ -H2AX at centromeres and found no change in the quantity of centromeric γ -H2AX across all APH treated cells regardless of TRF2 status (control, TRF2 shRNA or TRF2^{OE}). From these data we conclude that TRF2 function in mitotic death is exclusively telomeric. These data have been added as Supplementary Fig. 8g with corresponding text on page 15, middle paragraph.

6. Authors should refer to the following studies as they are pertinent to their manuscript.

-Haarhuis et al. 2013. Current Biology. WAPL-Mediated Removal of Cohesin Protects against Segregation Errors and Aneuploidy

-Mendez-Bermudez et al. 2018. Genome-wide Control of Heterochromatin Replication by the Telomere Capping Protein TRF2

We have cited both manuscripts as requested. They are now references 26 and 39.

Reviewer #3 (Remarks to the Author):

In this work, the authors investigate the cellular response to DNA replication stress. They employ Aphidicolin (APH) or Hydroxyurea (HU) to induce replication stress and visualize cellular outcome using live-cell imaging. They find that in p53-deficient cells, replication stress induces a spindle assembly checkpoint (SAC)-dependent mitotic arrest and cell death. They found that this mitotic death is triggered by the activation of two pathways: *BAX/BAK*-mediated apoptosis and a cohesion defect that is mediated by WAPL, and a secondary pathway triggered by telomere deprotection. They also provide evidence that suppressing mitotic death promotes genome instability in cells that experienced replication stress. The data in this study provide mechanistic insight into the pathways that upon replication stress induce mitotic catastrophe. This is useful information that may guide future experiments with implication for cancer treatment and defining the pathways that ensure genome stability.

1. The data in support of a relationship between suppression of mitotic cell death and induction of genome instability is not entirely convincing/clear. The authors provide few data in support of genome instability being an outcome of replication stress when mitotic cell death is inhibited. Moreover, the authors show that blocking mitotic death in two independent manners causes very different effects: WAPL inhibition, when combined with APH treatment, conferred an increase in multipolar cell division and micronuclei (Fig. 4j,k). In contrast, APH treatment in *BAX BAK* DKO cells did not result in the same effect. This is puzzling given that the authors concluded that WAPL and *BAX/BAK* act on the same

pathway (Supplemental Fig. 5). Additional experiments are required to clarify what are the consequences of replication stress when mitotic death is inhibited.

We agree with the reviewer's criticism regarding our assessment of genomic instability. A more appropriate description is that inhibiting mitotic death by WAPL depletion or *BAX BAK* DKO leads to different cellular outcomes. We have edited the abstract and text on page 12, 16, and 20 to appropriately convey our experimental assessment.

Please see our response to Reviewer #2, comment 4 regarding the different outcomes following APH treatment in WAPL depleted or *BAX BAK* DKO cells. WAPL and BAX/BAK both contribute to Type 1 mitotic death but do so through distinct functions in the pathway.

Our data are consistent with WAPL functioning upstream in replication stress conditions to potentiate mitotic arrest through the minor cohesion phenotype. Mitotic arrest leads to subsequent cohesion fatigue and ultimately *BAX/BAK*-dependent apoptosis. Inhibiting WAPL upstream rescues lethality and results in chromosome segregation errors and micronuclei. While inhibiting *BAX/BAK* downstream allows cohesion fatigue to progress in the absence of apoptosis. Eventually the *BAX/BAK* DKO cells escape mitotic arrest through slippage. This is supported by our new data showing the worsening of cohesion fatigue in APH treated *BAX BAK* DKO cells (Fig 5g, response to Reviewer #1 comment 1).

We did assess cell outcomes of mitotic slippage in APH treated *BAX BAK* DKO cells using higher resolution and higher magnification confocal microscopy. This revealed extensive multi-lobular nuclei in APH treated *BAX BAK* DKO cells (added as Supplementary Movie 4). Additionally, new data from experiments where cells were treated with APH in WAPL depleted or *BAX BAK* DKO cells, and then allowed to recover after APH washout, revealed the persistence of differing cellular outcomes following mitotic death escape (Figure 5h-j). These outcomes of escaping mitotic death are discussed correspondingly in the results on Page 12 and 13, and discussion on Page 17, 19 and 20.

REVIEWERS' COMMENTS:

Reviewer #1 (Remarks to the Author):

The authors have clarified most of the prior issues and added new data that make their models more convincing. This is a good contribution to the literature and worthy of publication in Nature Comm.

Reviewer #3 (Remarks to the Author):

The authors addressed all of my previous concerns. This manuscript represents a relevant body of work that helps to define the outcome of replication defects.